# OPENINS3D: SNAP AND LOOKUP FOR 3D OPEN-VOCABULARY INSTANCE SEGMENTATION

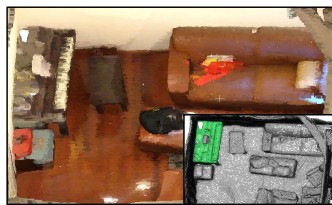 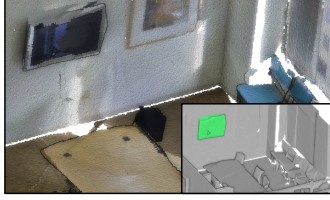 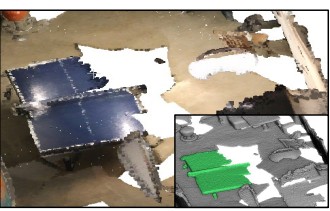

*"furniture that is capable of producing music"*     *"device to watch BBC news"*     *"Ma Long's domain of excellence "*

Figure 1: **Open-vocabulary Instance Segmentation Examples of OpenIns3D (LISA)**. OpenIns3D seamlessly transfers the open-world capability of 2D Vision and Language (VL) models into the 3D domain. LISA (Lai et al., 2023) is an LLM-based reasoning segmentation model.

## ABSTRACT

Current 3D open-vocabulary scene understanding methods mostly utilize well-aligned 2D images as the bridge to learn 3D features with language. However, applying these approaches becomes challenging in scenarios where 2D images are absent. In this work, we introduce a new pipeline, namely, *OpenIns3D*, which requires no 2D image inputs, for 3D open-vocabulary scene understanding at the instance level. The OpenIns3D framework employs a *"Mask-Snap-Lookup"* scheme. The *"Mask"* module learns class-agnostic mask proposals in 3D point clouds. The *"Snap"* module generates synthetic scene-level images at multiple scales and leverages 2D vision language models to extract interesting objects. The *"Lookup"* module searches through the outcomes of *"Snap"* with the help of Mask2Pixel maps, which contain the precise correspondence between 3D masks and synthetic images, to assign category names to the proposed masks. This 2D input-free and flexible approach achieves state-of-the-art results on a wide range of indoor and outdoor datasets by a large margin. Moreover, OpenIns3D allows for effortless switching of 2D detectors without re-training. When integrated with powerful 2D open-world models such as ODISE and GroundingDINO, excellent results were observed on open-vocabulary instance segmentation. When integrated with LLM-powered 2D models like LISA, it demonstrates a remarkable capacity to process highly complex text queries which require intricate reasoning and world knowledge. The code and model will be made publicly available.

## 1 INTRODUCTION

3D scene understanding plays a critical role in various domains, such as autonomous driving, robotic sensing, AR/VR, and manufacturing, among others. While the development of 3D closed-set understanding is relatively mature, scene understanding in an open-world setting is still in its infancy. Closed-set understanding can only handle a predefined set of concepts and scenarios and fails to provide valid responses when faced with unfamiliar concepts or variations in language usage. This limitation impacts its performance in dynamic and ever-changing contexts.

Thanks to internet-scale image-text datasets, significant progress has been made in 2D image open-world understanding (Radford et al., 2021; Zhou et al., 2022; Kirillov et al., 2023; Chen et al., 2023b; Xu et al., 2023; Brown et al., 2020; Ding et al., 2022; Zhou et al., 2021). However, unlike 2D data

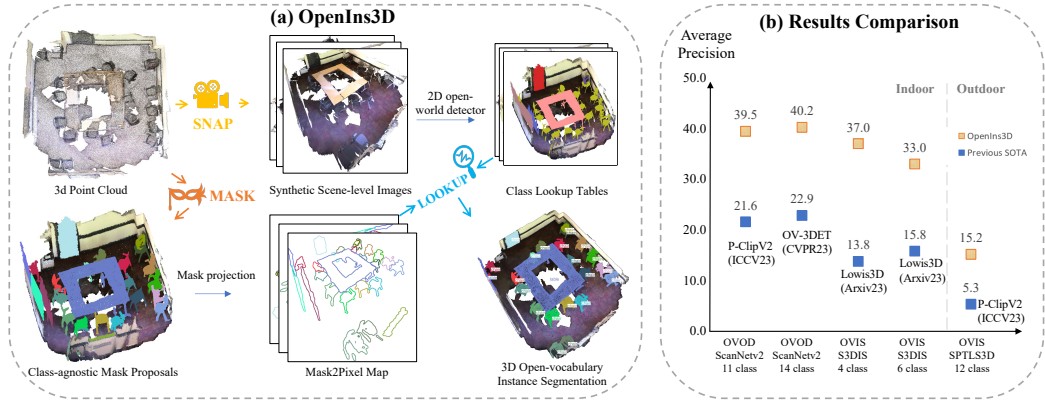

Figure 2: **High-level Illustrations of OpenIns3D and Quantitative Results**. (a) OpenIns3D follows the *"Mask-Snap-Lookup"* steps for open-vocabulary scene understanding. (b) A list of SOTA results has been achieved on both indoor and outdoor datasets. OVOD: open-vocabulary object detection. OVIS: open-vocabulary instance segmentation. P-CLIPV2 (Zhu et al., 2023); Lowis3D (Ding et al., 2023b); OV-3DET (Lu et al., 2023).

that can be easily collected from the internet, constructing a large-scale 3D-text dataset poses a challenge. As a result, the most viable approach to achieving 3D open-vocabulary understanding involves leveraging 2D images to bridge language and 3D data.

In this direction, there have been several notable works, such as OpenScene (Peng et al., 2023), PLA-family (Ding et al., 2023a; Yang et al., 2023a; Ding et al., 2023b), and CLIP2Scene (Chen et al., 2023a). These works leverage well-aligned 2D images and 3D point clouds to conduct feature distillation or employ 2D caption models to construct 3D-text pairs. One prerequisite of these methods, however, is the availability of well-aligned 2D images and 3D point clouds. This means that posed 2D images and associated depth maps need to be accessible as inputs to the network. For this reason, these methods are mostly applied to RGB-D-formed point clouds (Silberman et al., 2012; Dai et al., 2017; Rozenberszki et al., 2022; Chang et al., 2017). In real-life scenarios, there are numerous cases where meeting this prerequisite is challenging, either because 2D images are unavailable, or the information required to align 2D and 3D data is missing. For example, to save storage space, colored point clouds generated with LiDAR are often stored without accompanying 2D images. Example datasets include Hackel et al. (2017); Tan et al. (2020); Roynard et al. (2018). Similarly, point clouds produced by photogrammetry may be stored without depth maps, as seen in datasets like Hu et al. (2022); Chen et al. (2022); Li et al. (2020). In cases where point clouds are obtained from the registration of multiple scans from diverse sensors or are converted from 3D simulations/CAD models (Mo et al., 2019; Griffiths & Boehm, 2019), 2D images are often not generated. In the past, there have been attempts at 3D open-vocabulary understanding that do not require 2D input. However, these methods (Huang et al., 2023; Zhang et al., 2022; Zhu et al., 2023) primarily focus on object-level classification and face challenges in achieving satisfactory performance at the scene level.

To address this issue, we introduce **OpenIns3D**, a framework that exclusively utilizes colored **3D** point clouds as input to perform **Open**-vocabulary **Ins**tance understanding. Overall, OpenIns3D comprises three core steps: *Mask*, *Snap*, and *Lookup*. An overall illustration of OpenIns3D is presented in Figure 2a.

**Mask**: Given a 3D point cloud, the first part of OpenIns3D learns class-agnostic mask proposals with a *Mask Proposal Module* (MPM). This process is trained without any classification labels. To control the quality of the mask, MPM proposes a learnable *Mask Scoring* module to predict the quality of each mask output and implements a list of *Mask Filtering* techniques to discard invalid, low-quality masks. MPM outputs a list of class-agnostic masks in the scene.

**Snap**: Multiple synthetic scene-level images are generated with calibrated and optimized camera poses and intrinsic parameters. These images are specifically designed to encompass all relevant masks, aiming to minimize the need for multiple renderings. Instead of individually inferring each

mask (Zhu et al., 2023; Zhang et al., 2022; Lu et al., 2023), the scene-level images are input into 2D open-world models for the simultaneous understanding of interesting objects present in the scene. A *Class Lookup Table* (CLT) is then constructed to store all the detected object categories alongside their respective pixel locations.

**Lookup**: To precisely determine the positions of mask proposals in each image, Mask2Pixel maps are constructed. These maps project all 3D mask proposals onto 2D images with identical camera parameters used in *Snap*. In the *Lookup* phase, OpenIns3D searches through the CLT with the help of Mask2Pixel maps to precisely assign category names to 3D mask proposals. Results from multiple views are combined to establish initial mask classification outcomes. For remaining masks, a similar *Lookup* procedure is carried out on a local scale to facilitate classification. Lastly, the 3D mask proposals are refined by removing masks lacking class assignments after both global and local *Lookup*.

This simple and flexible approach has proven to be highly effective across a variety of indoor and outdoor benchmarks (Figure 2b). For indoor scenes, OpenIns3D was evaluated on the S3DIS (Armeni et al., 2016) and ScanNetv2 (Dai et al., 2017) datasets without utilizing any 2D images, pose information or depth maps. It was compared with other models, irrespective of their input prerequisites. For outdoor scenes, tests were conducted on an outdoor aerial photogrammetry dataset, STPLS3D (Chen et al., 2022). OpenIns3D achieved state-of-the-art results in open-vocabulary instance segmentation (OVIS) on both S3DIS (+> 17.2%) and STPLS3D (+> 10.1%). When converting mask proposals into 3D bounding boxes, OpenIns3D also achieved state-of-the-art results in open-vocabulary object detection (OVOD) on ScanNetv2 (+> 17.4%).

Additionally, the *Snap* and *Lookup* modules operate under a zero-shot scheme, allowing the 2D detector to be changed without the need for re-training. This confers a significant advantage to OpenIns3D, enabling it to seamlessly adapt to the latest 2D VL models. As a result, when integrated with robust 2D VL models like ODISE (Xu et al., 2023) and GroundingDINO (Liu et al., 2023), impressive segmentation results are achieved across various benchmarks. Moreover, when integrated with LISA (Lai et al., 2023), an LLM-powered reasoning segmentation model, OpenIns3D exhibits a strong capability to comprehend highly intricate language queries, including those requiring complex reasoning or world knowledge, as illustrated in Figure 1. Our contributions are:

- Unlike other 3D open-world methods, OpenIns3D employs a distinct pipeline that operates without the need for well-aligned images. This approach not only achieves state-of-the-art results across a range of benchmarks but also possesses the strong capability to comprehend highly complex input queries.

- We propose a *Mask Proposal Module* to effectively learn class-agnostic mask proposals and filter out low-quality outputs without the need for classification labels.

- We introduce a framework to optimize and calibrate the pose and intrinsic parameters of cameras to produce high-quality synthetic scene-level images from 3D point clouds, which proves more compatible with 2D VL models.

- We design a *Mask2Pixel Guided Lookup* in the *Lookup* module to seamlessly link 2D results with the 3D mask, proving to be very highly effective.

- Compared with other work that requires generating images from point clouds, our synthetic scene-level image approach not only requires less rendering time and inference time but also achieves much stronger results.

## 2 RELATED WORK

### 2.1 3D OPEN-VOCABULARY UNDERSTANDING

Progress in 3D open-vocabulary understanding has been relatively slow compared to that of images. In the domain of 3D object classification tasks, methods like PointCLIP (Zhang et al., 2022), Point-CLIPV2 (Zhu et al., 2023), and CLIP2Point (Huang et al., 2023) project 3D point clouds into depth maps and link them with 2D models for classification. However, these methods lack performance in scene-level understanding, where points are often overlapped and incomplete. For scene-level understanding, most work has primarily focused on leveraging well-aligned 2D posed images, depth maps, and point clouds (Chen et al., 2023a; Peng et al., 2023; Rozenberszki et al., 2022; Ding et al., 2023a; Yang et al., 2023a; Ding et al., 2023b; Zeng et al., 2023). One notable example is OpenScene (Peng et al., 2023), which takes posed 2D images, depth maps, and 3D data as input, and feature

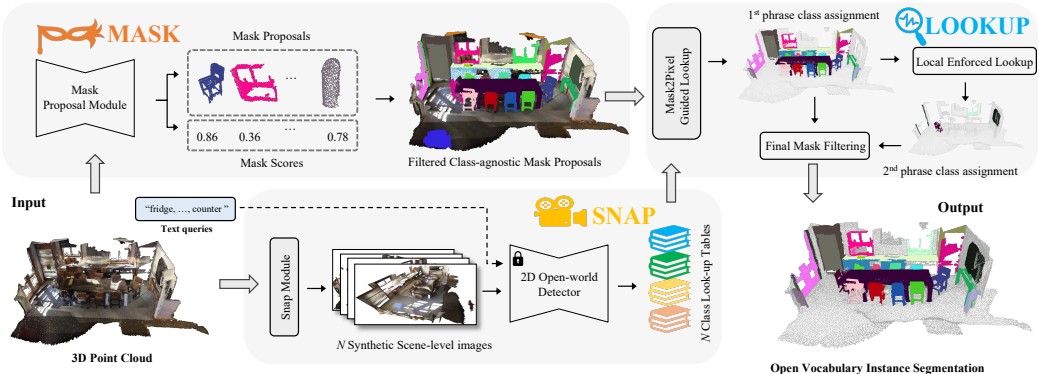

Figure 3: **General Pipeline of OpenIns3D Framework.** As a 3D-only framework, OpenIns3D first passes the point clouds into the MPM to generate both 3D mask proposals and mask scores. The *Snap* module (detailed in Figure 4) is then carried out to render $N$ synthetic scene-level images, which are later passed into the 2D open-world model along with the input text queries. The detection results from the 2D model are stored in *Class Lookup Table* (CLT). Finally, both the mask proposals and CLT are fed into the *Lookup* module, where *Mask2Pixel Guided Lookup* (detailed in Figure 5) is performed at the global level, followed by a *Local Enforced Lookup* at the local level to unlock the semantic meaning of mask proposals. The final mask filtering refines the mask proposals and obtains the final results.

distillation is performed to transfer 2D language-aligned features from images to 3D point clouds. Similarly, Clip2Scene (Chen et al., 2023a) builds dense pixel-point pairs by calibrating the LiDAR point cloud with corresponding images captured by six cameras. However, achieving instance-level understanding is challenging with these methods as they focus solely on semantic-level understanding. In contrast, PLA (Ding et al., 2023a) and its follow-up work RegionPLC (Yang et al., 2023a) and Lowis3D (Ding et al., 2023b) utilize a 2D caption model to construct 3D-text pairs to learn features. However, the PLA-family of works relies on a binary head to classify the input object into base-categories or novel-categories, and the transferability of this binary head to different base-novel splits is very limited, posing a challenge for flexible applications. A common issue with these methods is their reliance on well-aligned 3D and 2D pairs in the input, which may not always be available in real applications. Simplifying input requirements can enhance flexibility and compatibility, so exploring how to conduct open-vocabulary understanding without 2D images is an avenue worth pursuing.

## 2.2 IMAGE GENERATION FROM 3D

Projection-based methods have been extensively explored in the past for 3D understanding and have proven to be beneficial for obtaining complementary features. For instance, MVCNN (Su et al., 2015) projects 3D objects to different views to aid in feature learning, while LAR (Bakr et al., 2022) introduces object centre projection methods to generate images for 3D objects from various angles, assisting visual grounding tasks. Additionally, Virtual View Fusion (Kundu et al., 2020) employs the original camera pose but enlarges the field of view, resulting in enhanced 2D feature transfer. However, these methods encounter challenges like best view selection, object occlusion, information loss during projection, and long rendering times. In the context of open-vocabulary settings, the quality of the projected image plays a crucial role in model performance. In our work, we evaluate different projection methods, along with their compatibility with 2D open-vocabulary models, to identify an optimal solution that achieves good results and is efficient to implement.

## 3 BASELINE AND CHALLENGES

The general framework of OpenIns3D is inspired by recent unified image segmentation models (Cheng et al., 2022; Carion et al., 2020; Zhu et al., 2020), which follow a two-stage paradigm: binary mask extraction and mask classification. In our OpenIns3D, the *Mask Proposal Module* (MPM) focuses on the first task, the *Snap* and *Lookup* modules focus on the second task, and are combined to form the final instance segmentation results. We build a naive baseline of OpenIns3D by adopting the recent 3D instance segmentation backbone Mask3D (Schult et al., 2023) to generate

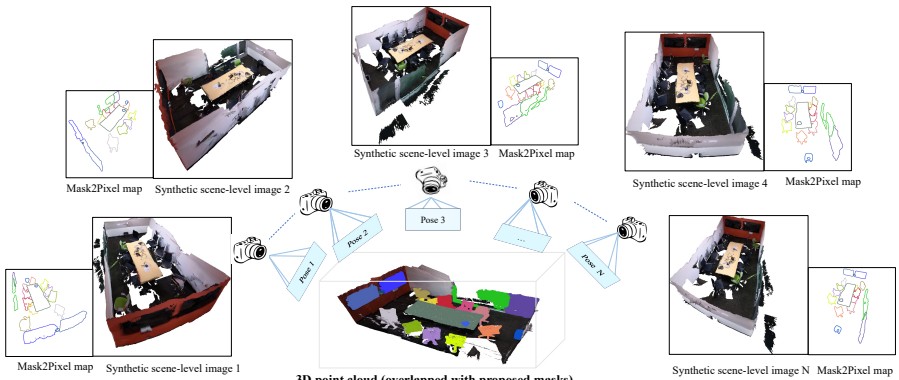

Figure 4: *Snap* and **Mask2Pixel Maps.** Cameras are positioned evenly at the outer boundary of the scene and elevated by 1m to capture a clear view. Each camera is pointed towards the scene centre. All images are calibrated to encompass all proposed masks. Pose and intrinsic matrix are stored in CLT, which is later used to generate the Mask2Pixel maps (using the same color to represent 2D-3D correspondences) to guide the search for category names.

mask proposals. To make MPM fit for the open-vocabulary setting, we remove all components in Mask3D that use the classification labels. Later, PointCLIP (Zhu et al., 2023) is adopted for mask understanding. This naive approach, although satisfying the requirement of 3D inputs only, has long rendering times and unsatisfactory performance (more details in Table 2 and Table 13). The issues of the baseline model can be summarized as follows:

**Challenge 1** *Excessive mask proposals.* The model generates a large number of low-quality mask proposals. Originally, mask proposals are filtered by the mask classification logit, which is removed in class-agnostic setting. Therefore, an effective mask filtering scheme is needed.

**Challenge 2** *Low quality of 3D instances.* 3D instances extracted from scene scans are typically broken, distorted, and sparse, posing challenges for rendering quality 2D images. As a result, the generated images are not easily understood by 2D VL models.

**Challenge 3** *Lack of context information.* As humans, our ability to recognize imperfect 3D point cloud object stems from the overall scene comprehension. However, individual mask point cloud projections lack this contextual information. A straightforward solution is to project not just the mask but also the background onto the images. However, this introduces additional distracting objects and irrelevant elements, potentially confusing classification-level VL models like CLIP.

**Challenge 4** *Domain gap between projected images and natural images.* It is challenging for a 2D VL model to understand rendered images, as they are very different from the images used in the training process, which poses a domain gap.

## 4 OPENINS3D

In this section, we present our design of OpenIns3D, which targets the aforementioned four challenges. The overall pipeline of OpenIns3D is shown in Figure 3.

### 4.1 MASK PROPOSAL MODULE

**Mask Scoring** Inspired by (Huang et al., 2019; Kirillov et al., 2023; Chen et al., 2023b), we propose a simple yet effective *Mask Scoring* design to eliminate low-quality masks in the inference process. Specifically, we feed the instance queries in the Mask Module in the 3D backbone to a shallow MLP module to predict the quality of the mask using IoU as the indicator. The predicted IoU ($IoU_m$) is supervised by the ground truth IoU ($IoU_{gt}$) value during the training stages, which is calculated between the predicted mask and its matched ground truth mask in the Bipartite Matching. For unmatched prediction masks ($IoU_u$), we label the ground truth IoU value as zero. $L2$ loss is used to compute the loss. To avoid overly low IoU predictions, a hyper-parameter $\gamma$ is introduced to reduce the weight of loss for unmatched masks. Therefore, the total loss function for the mask quality module is:

$$\mathcal{L}_{\text{total}} = \gamma \sum (\text{IoU}_u)^2 + \sum (\text{IoU}_m - \text{IoU}_{\text{gt}})^2 \tag{1}$$

**Mask Filtering**    To enhance mask quality, three filters are applied. Firstly, we retain masks with a model-predicted IoU score above a threshold of $\beta$, ensuring that only high-quality masks are kept. Secondly, drawing inspiration from SAM (Kirillov et al., 2023), we focus on stable masks by comparing two binary masks derived from the same underlying soft mask using different threshold values. Specifically, we introduce an offset value $\alpha$ and select masks where the IoU between the pair of thresholded masks (one with $-\alpha$ and the other with $+\alpha$) exceeds 80%. Lastly, small objects in the scene often lead to invalid proposals, so we filter out mask proposals that have a point number lower than $N_{\min}$. With these techniques, we significantly reduce the number of mask proposals (*Challenge 1*) and obtain cleaner and higher quality masks for subsequent mask understanding tasks.

## 4.2    SNAP: SYNTHETIC SCENE-LEVEL VIEW GENE

Rendering images from points can be a time-consuming task, especially when the number of rendering jobs is high. After exploring various techniques, we propose a synthetic scene-level image scheme that is not only highly efficient but also effective (*Challenges 2 and 3*). Details of attempts can be found in Sec 5.2 and Appendix E.

**Camera Extrinsic Parameter**    Cameras are strategically positioned above the scene to capture a clearer view. These camera positions are evenly distributed in a circular arrangement, ensuring images are captured from various perspectives (Figure 4). Each camera is oriented to point directly toward the centre of the scene. With the camera position coordinate $P_{cam}$, and target coordinate $P_{target}$, as well as the up axis of the scene $U$, we use the *Lookat* function to determine the pose matrix $Pose$. We elaborate on this method in Appendix A.2.

**Camera Intrinsic Calibration**    Once the camera extrinsic matrix is established, a camera intrinsic calibration is conducted, with the goal of encompassing the entire scene within the images. To achieve this, we initiate an arbitrary camera intrinsic matrix and

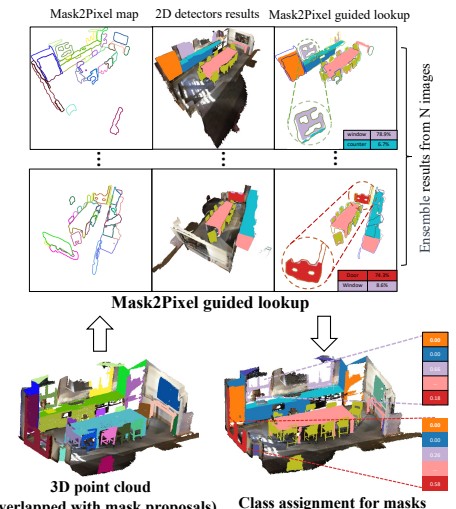

Figure 5: **Mask2Pixel Guided Lookup Illustration.** IoUs between the 2D detection results and the projected masks are the guidance to assign class names to 3D masks. Results from multiple images are ensembled for the final prediction.

then adjust the focal lengths ($f_x$ and $f_y$) and the central coordinates of the image ($c_x$ and $c_y$) through scaling. This ensures that the entire scene is captured within the image and maximizes the utilization of image pixels.

**Class Lookup Table**    Upon obtaining $N$ synthetic scene-level images, we input them into a 2D open-vocabulary detector. With text queries provided for interested classes, a list of detected objects in synthetic images can be obtained. Subsequently, information about detected objects, including their location and class, are stored in a designated *Class Lookup Table* (CLT). This table will later be retrieved to allocate class categories to 3D mask proposals. The primary objective of the 2D open-vocabulary detector is to identify as many objects as possible, and the structure of location information remains adaptable. This means that any 2D models for open-vocabulary segmentation or detection will be suitable (*Challenge 4*).

## 4.3    LOOKUP: MASK CLASSIFICATION THROUGH SEARCHING

**Mask2Pixel Guided Lookup**    For accurate searching within the CLT, we introduce a *Mask2Pixel Guided Lookup* (MGL). The concept involves projecting each 3D mask proposal onto a 2D plane using the same camera extrinsic and intrinsic matrices that are utilized to generate the 2D image, as depicted in Figure 4. With knowledge of the precise pixel locations of each mask in images, we

Table 1: **3D Open-vocabulary Instance Segmentation Results on S3DIS and ScanNetv2.** We compare our zero-shot performance on the same novel class splits with PLA-family works. Significant improvements are achieved on the S3DIS dataset, and competitive results are observed on ScanNetv2.

| Indoor. OVIS. | | S3DIS | | | ScanNetv2 | | |
|---|---|---|---|---|---|---|---|
| Method | B/N | AP50 | AP25 | B/N | AP50 | AP25 | use 2D |
| PLA | 8/4 | 08.6 | – | 10/7 | 21.9 | – | ✓ |
| RegionPLC | – | – | – | 10/7 | 32.3 | – | ✓ |
| Lowis3D | 8/4 | 13.8 | – | 10/7 | 31.2 | – | ✓ |
| Mask3D-P-CLIP | –/4 | 05.4 | 10.3 | –/7 | 04.5 | 07.8 | ✗ |
| OpenIns3D (ours) | –/4 | 37.0 (+23.2) | 39.3 (+29.0) | –/7 | 27.9 | 42.6 (+34.8) | ✗ |
| PLA | 6/6 | 09.8 | – | 8/9 | 25.1 | – | ✓ |
| RegionPLC | – | – | – | 8/9 | 32.2 | – | ✓ |
| Lowis3D | 6/6 | 15.8 | – | 8/9 | 38.1 | – | ✓ |
| Mask3D-P-CLIP | –/6 | 08.5 | 10.6 | –/9 | 05.6 | 06.7 | ✗ |
| OpenIns3D (ours) | –/6 | 33.0 (+17.2) | 38.9 (+28.3) | –/9 | 19.5 | 27.9 (+21.2) | ✗ |
| Mask3D-P-CLIP | –/12 | 08.6 | 09.3 | –/17 | 04.5 | 14.4 | ✗ |
| OpenIns3D (ours) | –/12 | 28.3 (+19.7) | 29.5 (+20.2) | –/17 | 28.7 (+24.2) | 38.9 (+24.5) | ✗ |

Table 2: **Rendering and Inference Time Ablations.** Results tested on scenes with 50 masks. OpenIns3D requires less rendering time, and inference time, and has a much stronger performance.

| Rendering | Img needed | 2D backbone | Img size $(w \times h)$ | $T_{render}$ (s/scene) | $T_{infer}$ (s/scene) | $T_{total}$ (s/scene) | AP25 (%) |
|---|---|---|---|---|---|---|---|
| PointCLIP | 250 | CLIP | $128^2$ | 5.2 | 15.3 | 20.5 | 9.3 |
| LAR | 250 | CLIP | $128^2$ | 14.3 | 18.7 | 33.0 | 10.5 |
| Mask rendering | 250 | CLIP | $128^2$ | 42.6 | 19.5 | 62.1 | 7.3 |
| OpenIns3D (ours) | 8 | G-DINO | $1000^2$ | 2.3 | 6.2 | 8.5 | 29.8 |
| OpenIns3D (ours) | 8 | ODISE | $1000^2$ | 2.3 | 8.2 | 10.5 | 35.1 |

can conduct an accurate search through the CLT to identify the most likely class for each mask. The development of MPM takes into account occlusion by integrating depth information. To accomplish the matching, we follow a three-step approach: 1. based on the mask's projection onto the 2D plane, we select the best-matched class categories in terms of IoU values; 2. if the IoU value of the best-matched object on the 2D plane is below 20%, the match is disregarded; 3. we aggregate results from multiple views to formulate the final prediction, calculating probability scores using their normalized average IoU values. This process is illustrated in Figure 5.

**Local Enforced Lookup**  While the *Mask2Pixel Guided Lookup* assigns class categories to mask proposals, some masks may not correspond to objects in the CLT. To address this, we introduce a *Local Enforced Lookup* (LEL) approach. We crop out the remaining masks from 2D scene-level images using enlarged bounding boxes and process them with the 2D detector to encourage detection. To select the best views, we introduce an *Occlusion Report* method (more details in Appendix A.4) to assess occlusion conditions for each mask in each projection, and then choose the top $K$ views for LEL.

**Final Mask Proposal Refinement**  With the previous lookup approaches, a large proportion of mask proposals obtain a category prediction. The final step of the network is to further refine the proposed masks in the MPM module. Specifically, all masks that have no category predictions after the MGL and LEL stages are eliminated.

## 5 EXPERIMENTS

**Datasets and Class Definition**  We tested OpenIns3D on three datasets that contain instance segmentation ground truth: S3DIS (Armeni et al., 2016), ScanNetv2 (Dai et al., 2017), and STPLS3D (Chen et al., 2022). S3DIS and ScanNetv2 are indoor point cloud datasets generated from RGB-D images, while STPLS3D is an aerial photogrammetry-constructed outdoor dataset. We exclusively utilized the 3D data from these datasets and did not employ any 2D images, poses, or depth maps.

In the closed-set instance segmentation, ScanNetv2 consists of 18 classes, S3DIS contains 13 classes, and STPLS3D contains 14 classes. While many other works randomly pick several classes

Table 3: **OVIS on STPLS3D.** OpenIns3D also works well on outdoor point clouds.

| Methods | AP50 | AP25 |
|---|---|---|
| Mask3D-P-CLIP | 02.6 | 04.0 |
| Mask3D-P-CLIPV2 | 03.1 | 05.2 |
| OpenIns3D (ours) | 13.3 (+10.2) | 15.3 (+10.1) |

Table 4: **Number of Views Ablation.** Tested on ScanNetv2 OVIS. LEL: Local Enforced Lookup.

| IDX | 4 | 8 | 16 | LEL | AP50 | AP25 |
|---|---|---|---|---|---|---|
| 1 | ✓ | | | | 18.3 | 27.1 |
| 2 | | ✓ | | | 22.7 | 35.1 |
| 3 | | | ✓ | | 24.8 | 37.5 |
| 4 | | | ✓ | ✓ | 28.7 | 38.9 |

Table 5: **CA-Mask Quality Evaluation.** Results tested on ScanNetv2 Validation set.

| Method | AP50 | AP25 |
|---|---|---|
| Mask3d-Supervised | 74.7 | 80.9 |
| CA-Mask3d | 47.5 | 49.2 |
| CA-Mask3d + MS | 50.2 (+02.7) | 53.3 (+04.1) |
| CA-Mask3d + MF | 61.6 (+14.1) | 71.0 (+21.8) |
| CA-Mask3d + MS + MF | 64.6 (+17.0) | 73.4 (+24.2) |

Table 6: **Cross-domain Ablations.** Results of models when trained and tested on different datasets.

| Test | Model | Training Data | AP50 | AP25 |
|---|---|---|---|---|
| ScanNet | Mask3D-P-CLIP | ScanNetv2 | 04.5 | 14.4 |
| | OpenIns3D | ScanNet | 28.7 | 38.9 |
| | OpenIns3D | S3DIS | 21.5 | 33.6 |
| S3DIS | Mask3D-P-CLIP | S3DIS | 03.5 | 06.8 |
| | OpenIns3D | S3DIS | 28.3 | 29.5 |
| | OpenIns3D | ScanNetv2 | 14.2 | 19.8 |

for evaluation, we used the following scheme to closely follow the closed-set setting. We used on the categories setting of PLA, excluding the "other furniture" class in ScanNetv2 and the "clutter" class in S3DIS due to their vague meanings. For STPLS3D, we merged the low, medium, and high vegetation classes into one "vegetation" class and kept all the rest.

**Adapted Metrics for Comparison with SOTA** We adopted various comparison schemes to align with existing methods. For 3D Open-vocabulary Instance Segmentation, we compare with PLA (Ding et al., 2023a), and its follow-up works RegionPLC (Yang et al., 2023a) and Lowis3D (Ding et al., 2023b). Unlike the zero-shot inference of OpenIns3D, these three works are trained on base categories and tested on novel categories. For a fair comparison, we followed their category splits and compared our results on novel classes, as demonstrated in Table 1. For STPLS3D, we compared OpenIns3D with baseline models whose classification module is PointCLIP and PointCLIPV2 (Zhu et al., 2023) (Table 3).

For 3D Open-vocabulary Object Detection, we compared our method with PointCLIP (Zhang et al., 2022) and PointCLIPV2 (Zhu et al., 2023), as well as OV-3DET (Lu et al., 2023). PointCLIP and PointCLIPV2 use pre-trained 3DETR (Misra et al., 2021) to generate bounding box proposals, while OV-3DET utilizes posed 2D images and pre-trained 2D DETR (Carion et al., 2020) to generate bounding box proposals. OpenIns3D's MPM was trained from **scratch** and used no classification labels. Note that these OVOD methods are evaluated only on some categories. For a fair comparison, Table 7 lists our per-category results and compares them fairly with their settings.

## 5.1 COMPARISON WITH SOTA

For 3D instance segmentation, compared to works in the PLA family (Ding et al., 2023b; Yang et al., 2023a; Ding et al., 2023b), OpenIns3D does not require assistance from original images or training on base categories. It still achieves significantly higher results on the S3DIS dataset, both in the 4 novel categories split and the 6 novel categories split, surpassing the previous state-of-the-art by 23.2% and 17.2%, respectively. In ScanNetv2, OpenIns3D demonstrates competitive performance. In SPTLS3D, OpenIns3D outperforms the baseline model PointCLIPV2 by 10.2% and 10.1% in AP50 and AP25, respectively. For 3D object detection, OpenIns3D showcases excellent performance, outperforming all previous methods by more than 17%. (More details in Appendix D)

## 5.2 ABLATION STUDY

**Mask Quality Ablation** Following the evaluation on ScanNetv2, we assessed the class-agnostic mask quality using the Average Precision score. We treated all classes as universal since the predictions are class-agnostic. The evaluation was conducted on the ScanNetv2 Validation set. Table 5 demonstrates the effectiveness of the *Mask Scoring* and *Mask Filtering* designs.

Table 7: **3D Open-vocabulary Object Detection Results on ScanNetv2, in AP25.** P-3DE: pre-trained 3DETR (Misra et al., 2021). P-2DET : pre-trained DETR (Carion et al., 2020) + projection. MPM: 3D box from MPM.

| Methods | BBox | 11-class | 14-class | all-class | cab | bed | chair | sofa | table | door | cntr | desk | sink | bath | win | bkshf | cur | fri | toi | pic | shower |
|---|---|---|---|---|---|---|---|---|---|---|---|---|---|---|---|---|---|---|---|---|---|
| PointCLIP | P-3DE | 9.1 | – | – | 6.0 | 4.8 | 45.2 | 4.8 | 7.4 | 4.6 | 1.0 | 4.0 | 13.4 | 6.5 | 2.2 | – | – | – | – | – | – |
| PointCLIPV2 | P-3DE | 21.6 | – | – | 19.3 | 21.0 | 61.9 | 15.6 | 23.8 | 13.2 | 12.4 | 21.4 | 14.5 | 16.8 | 17.4 | – | – | – | – | – | – |
| OV-3DET | P-2DE | – | 22.8 | – | 3.0 | 42.3 | 27.1 | 31.5 | 14.2 | 9.6 | 0.3 | 19.7 | 31.6 | 56.3 | – | 5.6 | 10.5 | 11.0 | 57.3 | – | – |
| OIS3D (ours) | MPM | 39.5(+17.9) | 40.2(+17.4) | 36.0 | 17.1 | 57.5 | 74.5 | 59.2 | 36.9 | 29.3 | 31.1 | 32.2 | 42.1 | 6.6 | 47.5 | 26.4 | 55.4 | 39.1 | 57.4 | 0.0 | 0.0 |

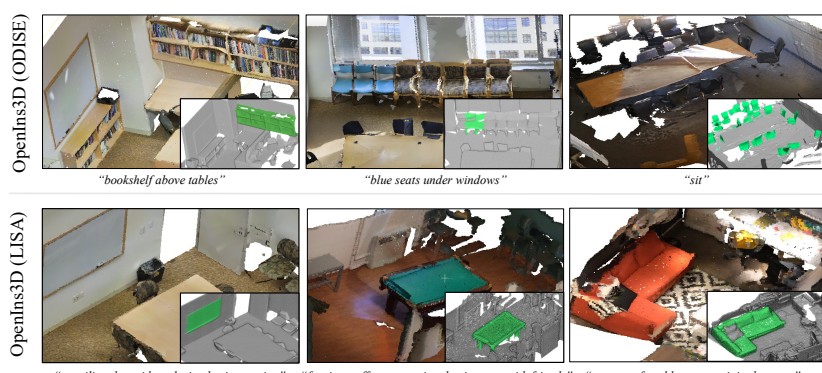

Figure 6: **Qualitative Results from OpenIns3D.** OpenIns3D (ODISE) demonstrates the ability to manage a versatile vocabulary. OpenIns3D (LISA) can conduct 3D reasoning segmentation.

**Multi-view Ablation** We also studied the effects of using different numbers of views (Table 4). Increasing the number of views used in the *Lookup* module leads to better results. Additionally, LEL provided a final boost to the results.

**Projection and 2D Backbone Ablation** Before delving into scene-level synthetic view generation, we conducted a comprehensive study on various rendering methods and their interaction with the 2D backbone to identify a suitable approach. We report the rendering time and inference performance for each method (Table 2). OpenIns3D not only excels in terms of rendering time but also demonstrates strong performance across all evaluated methods.

**Cross-domain Analysis** In this analysis, OpenIns3D was trained and tested on different datasets to examine its generalization capability, as shown in Table 6. The cross-domain models also demonstrate impressive performance on both datasets when compared with the baseline. Notably, within the 17 classes of ScanNetv2, 11 classes do not exist in S3DIS. OpenIns3D, trained on S3DIS, still achieves decent performance among these unseen classes (more details in Appendix D.2).

**Free-flow Language Capability** OpenIns3D can seamlessly transfer open-world capability in the 2D model to the 3D domain. When integrating OpenIns3D with a 2D model powered by LLM, such as LISA (Lai et al., 2023), OpenIns3D can handle highly sophisticated, abstract inputs that require prior knowledge of the world or complex reasoning (as shown in Figure 6).

## 6 CONCLUSION

Achieving 3D open-world scene understanding is a challenging task, primarily due to the lack of extensive 3D-text data. Currently, most work in this domain focuses on using 2D images to bridge the gap between 3D and language. This, however, not only requires a good alignment between 2D and 3D but also evolves slowly due to the significant effort needed for retraining when changing the 2D backbone. In contrast, OpenIns3D introduces a completely new pipeline, i.e. Mask-Snap-Lookup, for this task. The **Mask** module generates authentic masks in the 3D domain, while **Snap** renders scene-level images in 2D domains, and the **Lookup** module links the results from 2D to 3D precisely. This pipeline requires no 2D input (i.e. is more flexible), achieves very stronger performance (i.e. is more powerful) and can evolve seamlessly with a 2D model without training (i.e. is more fast-evolving). We hope our work will provide a fresh perspective for researchers working towards open-world 3D scene understanding and set next-level benchmarks for various tasks in the domain.

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

APPENDIX

OpenIns3D is a powerful, 2D input-free, fast-evolving, complex-input-handling framework for 3D open-world scene understanding. We showcase the differences between OpenIns3D and other frameworks in Figure 7.

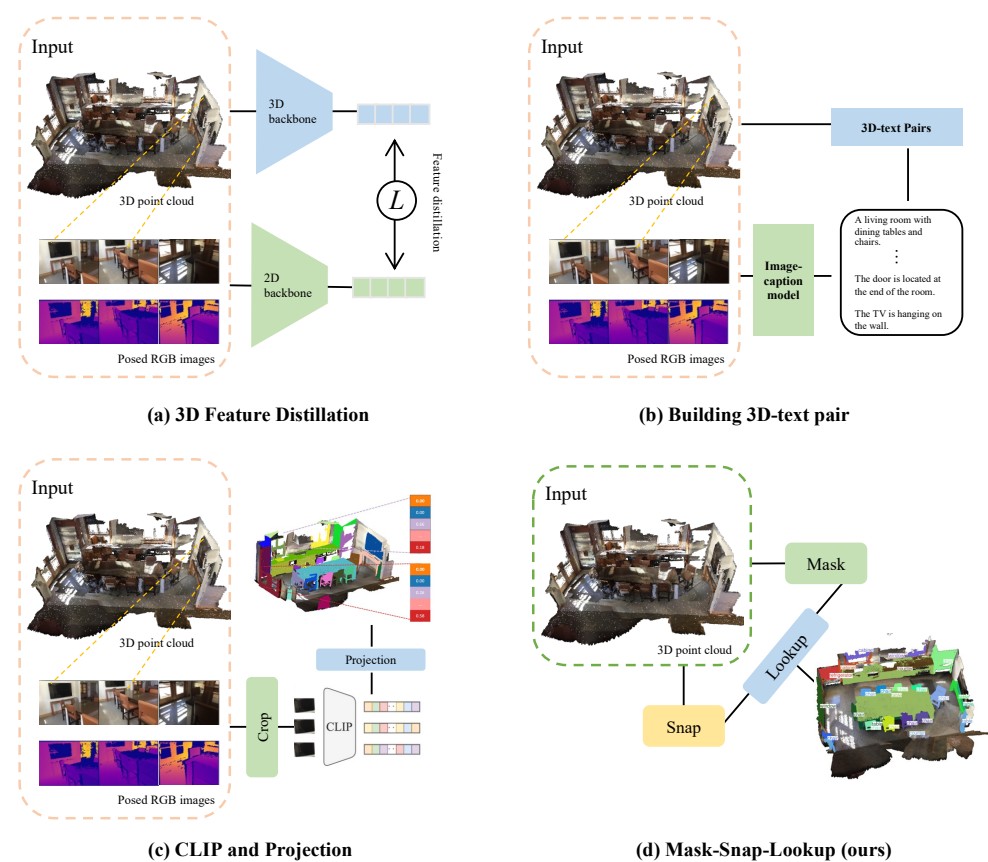

Figure 7: **Comparison of the OpenIns3D framework with other models.** OpenIns3D offers a unique capability of using only 3D input, making it more applicable in real-life scenarios. a) 3D feature distillation frameworks, where 2D images are used as a bridge to distill language-aligned features into 3D, with typical works including OpenScene (Peng et al., 2023) and Clip2Scene (Chen et al., 2023a). b) Building 3D-text pairs, where 2D captioning models are used to build 3D-text pairs for feature learning, with typical works including PLA-family (Ding et al., 2023b; Yang et al., 2023a; Ding et al., 2023a). c) CLIP and Projection, where objects are cropped out of 2D images before being processed by CLIP, and the results are directly projected into 3D, including OpenMask3D (Takmaz et al., 2023), OV-3DET (Lu et al., 2023) and CLIP² (Zeng et al., 2023). d) Mask-Snap-lookup, where only 3D input is needed for 3d open world scene understanding tasks

## A    MORE DETAILS ON METHODOLOGIES

### A.1    CLASS-AGNOSTIC MASK PROPOSAL MODULE

We modified components in Mask3D Schult et al. (2023) that require classification labels to make it a class-agnostic setting. This includes removing semantic probability components in Hungarian Matching, eliminating semantic classification loss, discarding classification logits-based ranking, and getting rid of classification logits-based filtering. Instead, we added the *Mask Scoring* module

and *Mask Filtering* techniques to acquire high-quality mask proposals without relying on semantic labels.

*Mask scoring* firstly utilizing the Hungarian Match to pair $N$ proposed masks with $n$ ground truth masks and calculating the IoU values. For $N - n$ unmatched masks, the IoU is set to zero. This yields GT IoUs for all proposed masks. Training involves using a two-layer MLP to process $N$ mask queries and predict their IoU values. The training is supervised by the the difference between predicted IoU and Ground Truth IoU, as shown in formula 1.

## A.2 CAMERA POSE GENERATION WITH *Lookat* FUNCTION

Here we detail how the pose matrix $Pose$ can be obtained using the *Lookat* function, followed by Falcidieno et al. (1992).

Given the camera position coordinates $P_{\text{cam}}$, which are located even at the top of the scene, and the camera target coordinate $P_{\text{target}}$, which is always the centre of the scene, along with the up axis of the scene $U$ (*i.e.* $[0, 0, -1]$), the pose matrix $Pose$ can be obtained as follows:

$$Pose = \begin{bmatrix} right_x & up_x & -forward_x & T_x \\ right_y & up_y & -forward_y & T_y \\ right_z & up_z & -forward_z & T_z \\ 0 & 0 & 0 & 1 \end{bmatrix}$$

Following the convention, "right" corresponds to the positive x-axis, "up" corresponds to the positive y-axis, and "forward" corresponds to the negative z-axis.

The normalized forward vector is the negative normalized direction from $P_{\text{cam}}$ to $P_{\text{target}}$:

$$forward = \frac{P_{\text{cam}} - P_{\text{target}}}{\|P_{\text{cam}} - P_{\text{target}}\|}$$

The normalized right vector is the *cross-product* between the up axis $U$ and the forward vector:

$$right = \frac{U \times forward}{\|U \times forward\|}$$

The normalized up vector is the *cross-product* between the forward vector and the right vector:

$$up = \frac{forward \times right}{\|forward \times right\|}$$

The translation values $T_x$, $T_y$, and $T_z$ are simply the components of the camera position $P_{\text{cam}}$:

$$T_x = P_{\text{cam}_x}, \quad T_y = P_{\text{cam}_y}, \quad T_z = P_{\text{cam}_z}$$

Finally, the $Pose$ matrix can be obtained by assembling these values into the $4 \times 4$ matrix format.

## A.3 CAMERA INTRINSIC CALIBRATION

Once the $Pose$ matrix is obtained, we initialize the intrinsic matrix with a standard intrinsic matrix $Intri$. Using both the $Pose$ and $Intri$, we perform a rapid point projection with the completed camera model, resulting in randomly positioned 2D scene. Subsequently, we uniformly scale the values of $fx$, $fy$, $cx$, and $cy$ in the initialized intrinsic matrix $Intri$ by the same factor to reposition and rescale the projected image into the center of the image plane. For example, if the original projected point was located in image coordinates within the range of $[-1000, -192]$ in $x$, our calibrated intrinsic metrics transform it to $[0, 2000]$ in $x$. Crucially, we preserve the ratio between $x$ and $y$ coordinates to achieve the final image without any additional loss in proportion. This procedure ensures that the utilization of each image is extensive and encompasses all the proposed masks.

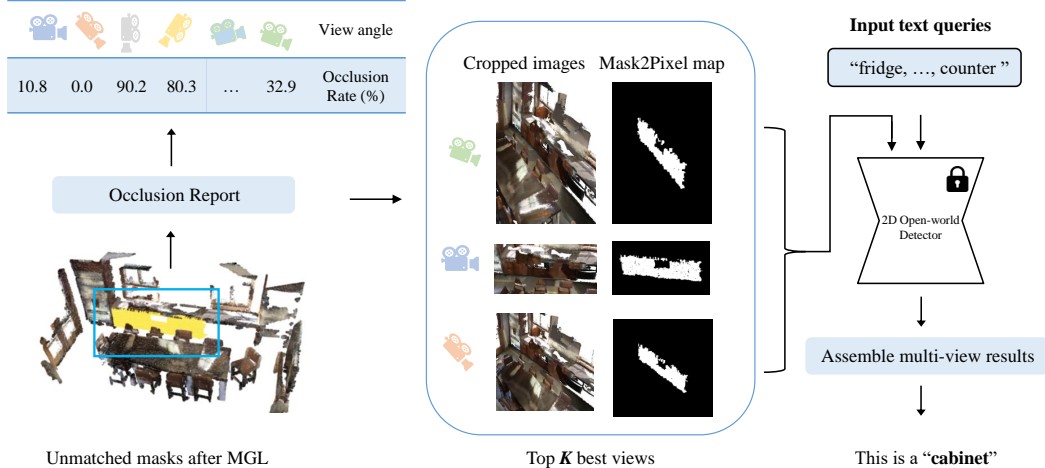

Figure 8: **Illustration of Local Enforced Lookup.** The remaining masks from phase one first go through the *Occlusion Report* module to select the best $K$ views. The selected images are cropped before being processed by the 2D detectors to encourage a classification result.

## A.4   LOCAL ENFORCED LOOKUP

Here, we provide a detailed explanation of the *Occlusion Report* module that we proposed to effectively evaluate the occlusion condition of masks in all synthetic images. Specifically, the following four steps are executed:

- **Step 1.  Point Count Array:**  We initiate the process by constructing a 3D array with dimensions $W \times H \times (M + 1)$, where $M$ represents the number of masks, and 1 is added to account for the background points. This array will be denoted as $PC$, *i.e.* point count, as it is designed to store the number of points of the 3D mask projected onto each pixel in the images. For example, if the pixel at coordinates $i, j$ is occupied by two points from the 3D mask $k$ during the projection, $PT_{i,j,k}$ will be assigned the value 2.

- **Step 2.  Foremost Point Identification:**  Utilizing the depth map generated during the projection process, we construct a 2D array named $FP$ with dimensions $W \times H$, which is used to identify the foremost point in each pixel and indicate the originating mask number. For example, if pixel i,j's foremost point is projected from Mask $k$, we denote $FP_{i,j} = k$.

- **Step 3.  Occlusion Rate Calculation:**  To evaluate the occlusion rate $(OR)$ for mask $k$ within specific images, we compute the following formula:

$$OR_k = \frac{\sum_{i=1}^{W} \sum_{j=1}^{H} PC_{i,j,k} \cdot (FP_{i,j} = k)}{T_k}$$

  where $T$ represent the total number of point in mask $k$.

- **Step 4. All Images Report:**  Finally, we repeat steps 1-3 for all images to obtain an overall report of the occlusion rate of each mask across all images, forming the final *Occlusion Report*.

After selecting the best view, synthetic scene-level images are cropped to focus on a specific mask proposal and then reprocessed by 2D detectors. The results are also searched with the help of Mask2Pixel maps to form the final classification prediction for the mask, as shown in Figure 8.

## B    IMPLEMENTATION DETAILS

### B.1    MASK

The Mask Proposal Module is built upon a lightweight version of Mask3D Schult et al. (2023) with 3 decoder layers. During training, we used 100 non-parameter queries for mask proposals. The bipartite matching process relies solely on the focal loss (weighted by 5) and dice loss (weighted by 2), without incorporating classification results. Despite not utilizing classification information, the Mask Proposal module is still capable of generating high-quality results. This is attributed to the diverse spatial distribution of 3D points, where two losses based on spatial information alone are sufficient for effective matching. For the mask quality scoring module, we set $\lambda$ to 0.1 to down-weight zero IOU masks.

The Mask Proposal module is trained using the ADAM optimizer with a learning rate of 0.0003, and the one-cycle scheduler is applied. For ScanNetv2, we follow the downsampling approach described in Wu et al. (2022), voxelizing the original input with a resolution of 0.02. We apply a series of augmentations, including flipping, elastic distortion, random rotation, chromatic auto-contrast, chromatic jitter, and translation. For S3DIS, we adopt the same settings as described in Schult et al. (2023), training the MPM on Areas 1, 2, 3, 4, and 6, and testing on Area 5. For STPLS3D, we split the scene into 50-meter spans and use preprocessing steps as outlined in Schult et al. (2023). We follow the training and validation split on STPLS3D and evaluate the performance of the validation set. All datasets are trained for 600 epochs on a single Nvidia A100 80G GPU.

### B.2    SNAP

We captured 16 images of the scene, evenly distributed along its outer boundary and focused on the centre of the scene. For all three datasets, we capture images with dimensions of 1000 x 1000 for a great trade-off between speed and performance. Additionally, to avoid the occlusion effect caused by the ceiling, we discard the top 0.3m points in the STPLS3D and ScanNetv2 datasets. As a result, the ceiling categories in the S3DIS dataset are completely discarded. We assign it a value of 0 in all matrices when evaluating and comparing with other methods. For STPLS3D, the camera position is located 5m higher than the top of the scene to acquire a better view.

### B.3    LOOKUP

During the *Lookup* stage, we only assign a classification label to each mask if the results have been verified in at least two views. This approach ensures a higher level of confidence in the assigned class labels. In the case of *Local Enforced Lookup*, we crop the images using bounding boxes that are twice the size of the target masks. The results are then fed into 2D detectors to refine the results. Mask2Pixel maps, in this case, binary maps, are used to accurately search for the detection results, as shown in Figure 8.

## C    EXPERIMENT ON SCANNET200

We further evaluate OpenIns3D's performance on a more challenging dataset ScanNet200 (Rozenberszki et al., 2022), which features a larger vocabulary and more categories. ScanNet200 comprises 200 classes, and based on the frequency of labeled points in the training set, these 200 classes are split into three groups: "head," which contains 66 categories; "common," which contains 68 categories; and "tail," which contains 66 categories. We report the results for each category groups.

While most other OpenScene models require 2D images during inference, the OpenScene 3D distillation model can process 3D point clouds directly for Open world understanding, making it also a 2d-input free model. Compared with this model, OpenIns3D demonstrates stronger performance across all category groups, with a notable 5.4% improvement, in head group.

In comparison to methods that use 2D aligned images during inference, OpenIns3D performs competitively on the head categories, yet its performance drops in the common and tail categories. This decline stems from OpenIns3D's sole reliance on the input 3D reconstruction. Within this

reconstruction, visual information for many small objects, particularly those in the common and tail groups, are likely to be diluted or lost, resulting in less optimal performance on these categories.

On the other hand, OpenMask3D leverages original 2D image for mask understanding, showcasing robust performance across all categories. However, this enhanced performance comes at the cost of incorporation of additional modality, leading to a reduction in flexibility in its application.

Table 8: **3D instance segmentation results on the ScanNet200 validation set.** OpenIns3D demonstrates robust performance, when compared to 2D-input free models. In comparison with models utilizing 2D images, it maintains competitive performance within the head categories split. However, notable limitations emerge when dealing with small objects in the common and tail classes.

| Model | Image Features | use 2D | head (AP) | common (AP) | tail (AP) | AP | $AP_{50}$ | $AP_{25}$ |
|---|---|---|---|---|---|---|---|---|
| OpenScene (2D Fusion) + masks | OpenSeg | ✓ | 13.4 | 11.6 | 9.9 | 11.7 | 15.2 | 17.8 |
| OpenScene (2D/3D Ens.) + masks | OpenSeg | ✓ | 11.0 | 3.2 | 1.1 | 5.3 | 6.7 | 8.1 |
| OpenScene (2D Fusion) + masks | LSeg | ✓ | 14.5 | 2.5 | 1.1 | 6.0 | 7.7 | 8.5 |
| OpenMask3D | CLIP | ✓ | 17.1 | 14.1 | 14.9 | 15.4 | 19.9 | 23.1 |
| OpenScene (3D Distill) + masks | OpenSeg | ✗ | 10.6 | 2.6 | 0.7 | 4.8 | 6.2 | 7.2 |
| OpenIns3D | ODISE | ✗ | **16.0** | **6.5** | **4.2** | **8.8** | **10.3** | **14.4** |

# D  PER CATEGORIES ANALYSIS

## D.1  COMPARISON WITH SOTA

Table 9 and Table 10 provide the per-class results of the proposed OpenIns3D on the S3DIS and ScanNetv2 datasets. We follow the performance of PLA and highlight the novel (unseen) classes. Note per categories results for RegionPLA and Lowis3D are not available. Table 11 represents the per-categories results for STPLS3D data, compared with PointCLIP and PointCLIPV2.

Table 9: **Per-class Results of 3D Open-vocabulary Instance Segmentation on S3DIS AP50.** Performance on novel classes is marked in blue.

| Methods | Partition | ceiling | floor | wall | beam | column | window | door | table | chair | sofa | bookcase | board |
|---|---|---|---|---|---|---|---|---|---|---|---|---|---|
| PLA Ding et al. (2023a) | B8/N4 | 89.5 | 100.0 | 50.8 | 00.0 | 35.3 | 36.2 | 60.5 | 00.1 | 84.6 | 01.9 | 00.8 | 59.4 |
| | B6/N6 | 89.5 | 60.2 | 17.9 | 00.0 | 41.5 | 10.2 | 02.1 | 00.6 | 86.2 | 45.1 | 00.1 | 02.2 |
| OpenIns3D | –/N12 | 00.0 | 84.4 | 29.0 | 00.0 | 00.0 | 62.6 | 25.2 | 25.5 | 52.0 | 60.0 | 00.0 | 00.0 |

Table 10: **Per-class Results of 3D Open-vocabulary Instance Segmentation on ScanNet AP50.** Performance on novel classes is marked in blue.

| Methods | Partition | cabinet | bed | chair | sofa | table | door | window | bookshelf | picture | counter | desk | curtain | fridge | shower c. | toilet | sink | bathtub |
|---|---|---|---|---|---|---|---|---|---|---|---|---|---|---|---|---|---|---|
| | B13/N4 | 50.5 | 77.0 | 82.9 | 43.4 | 75.4 | 49.0 | 46.0 | 43.7 | 46.5 | 33.7 | 23.2 | 54.1 | 49.6 | 56.0 | 97.8 | 47.5 | 85.8 |
| PLA Ding et al. (2023a) | B10/N7 | 53.7 | 62.7 | 11.2 | 70.5 | 27.2 | 47.7 | 45.7 | 30.0 | 01.5 | 39.9 | 40.8 | 50.6 | 68.6 | 84.6 | 92.9 | 24.6 | 00.0 |
| | B8/N9 | 45.1 | 77.4 | 82.2 | 84.2 | 74.2 | 48.9 | 51.0 | 30.0 | 00.5 | 02.1 | 16.8 | 44.9 | 28.3 | 35.1 | 94.3 | 16.6 | 00.0 |
| OpenIns3D | –/N17 | 24.3 | 52.5 | 75.7 | 61.6 | 40.6 | 39.7 | 45.5 | 54.8 | 0.5 | 33.5 | 16.7 | 48.1 | 18.5 | 4.3 | 50.1 | 16.8 | 7.6 |

Table 11: **Per-class Results of 3D Open-vocabulary Instance Segmentation on STPLS3D AP50.** All models are tested in a zero-shot manner.

| Methods | building | veg | vehicle | truck | aircraft | mil-veh | bike | motorbike | lightpole | signs | clutter | fence |
|---|---|---|---|---|---|---|---|---|---|---|---|---|
| PointCLIP | 15.3 | 0.4 | 10.2 | 06.6 | 00.0 | 00.0 | 00.0 | 00.0 | 00.0 | 00.0 | 00.0 | 00.0 |
| PointCLIPV2 | 20.3 | 0.2 | 12.3 | 5.8 | 00.0 | 00.0 | 00.0 | 00.0 | 00.0 | 00.0 | 00.0 | 00.0 |
| OpenIns3D | **40.4** | **01.2** | **54.2** | **24.2** | **30.0** | **05.5** | **02.1** | **03.0** | 00.0 | 00.0 | 00.0 | **08.3** |

In S3DIS, OpenIns3D consistently achieves high results for novel classes. We attribute this to the high quality of 3D point data in S3DIS, which ensures favourable conditions for recognition in Snap images.

However, for classes like columns, OpenIns3D struggles to produce meaningful results. Our performance on categories such as windows, floors, doors, tables, and sofas is typically at least 20% higher than PLA results. It is worth noting that PLA is partially trained on the base class and requires 2D images for captioning purposes.

For the ScanNetv2 dataset, our model performs better than PLA in certain categories such as chairs, sofas, tables, bookshelves, pictures, counters, and bathtubs. However, it slightly underperforms PLA in categories like beds, fridges, shower curtains, toilets, and sinks. In ScanNetv2, the quality of the point cloud data is not very high, especially for scans with higher scene IDs. As a result, the quality of Snap output is limited by the original point cloud, leading to slightly lower performance. Nevertheless, OpenIns3D, as a 2D input-free and label-free scheme, still achieves competitive performance on the ScanNetv2 dataset.

In the case of STPLS3D, our model outperforms PointCLIP and PointCLIPV2 by a significant margin in almost every category, achieving very high results, particularly in categories such as buildings, vehicles, trucks, and aircraft. However, the performance on very small objects, such as bikes, motorbikes, signs, and light poles, is not as strong. This is because the Snap module positions the camera at a high-level point to capture point cloud data from buildings, resulting in a limited number of pixels available for these smaller objects. This presents a challenge for OpenIns3D.

## D.2 Cross-domain Analysis

Table 12 presents the per-category results for the cross-domain OpenIns3D model, trained on S3DIS and tested on ScanNetv2. Despite the performance being relatively lower than that of the in-domain model (trained and tested on the same dataset), the performance is still competitive when compared to other SOTA results. Note that these SOTA models use pre-trained 2D/3D models to propose bounding boxes, which are trained with in-domain data (ScanNetv2)

The class categories between S3DIS and ScanNetv2 are very different. Within the 17 classes in ScanNetv2, only 6 classes exist in S3DIS. However, OpenIns3D, trained on S3DIS, can still perform relatively well in other categories that have never been seen before, demonstrating its good generalization capability.

This outcome is as anticipated, as the design of MPM closely resembles that of SAM in the 2D context, which has shown remarkable capabilities in mask proposal generation after being trained on a substantial amount of data. We believe that with more 3D class-agnostic labels available, the MPM module is capable of generating higher-quality Class-agnostic mask proposals.

Table 12: **Cross-domain Analysis of OpenIns3D on OVOD on ScanNetv2 AP25.** OpenIns3D achieves competitive results on the cross-domain dataset, even on categories at are not available on the training dataset, highlighted in blue . Compared with other SOTA models on OVOD, cross-domain OpenIns3D still has competitive performance. MPM-SC: MPM trained on ScanNetv2; MPM-S3: MPM trained on S3DIS.

| Methods | BBox Prop | cabinet | bed | chair | sofa | table | door | window | bookshelf | picture | counter | desk | curtain | fridge | shower c. | toilet | sink | bathtub |
|---------|-----------|---------|-----|-------|------|-------|------|--------|-----------|---------|---------|------|---------|--------|-----------|--------|------|---------|
| OpenIns3D | MPM-SC | 17.1 | 57.5 | 74.5 | 59.2 | 36.9 | 29.3 | 47.5 | 26.4 | 0.0 | 31.1 | 32.2 | 55.4 | 39.1 | 0.0 | 57.4 | 42.1 | 6.6 |
| OpenIns3D | MPM-S3 | 16.1 | 43.5 | 45.7 | 41.8 | 28.6 | 17.7 | 18.3 | 31.9 | 1.2 | 1.0 | 29.3 | 23.1 | 20.1 | 8.0 | 63.6 | 16.4 | 1.7 |
| *SOTA models* | | | | | | | | | | | | | | | | | | |
| PointCLIP | P-3DE | 6.0 | 4.8 | 45.2 | 4.8 | 7.4 | 4.6 | 2.2 | - | - | 1.0 | 4.0 | - | - | - | - | 13.4 | 6.5 |
| PointCLIPV2 | P-3DE | 19.3 | 21.0 | 61.9 | 15.6 | 23.8 | 13.2 | 17.4 | - | - | 12.4 | 21.4 | - | - | - | - | 14.5 | 16.8 |
| OV-3DET | P-2DE | 3.0 | 42.3 | 27.1 | 31.5 | 14.2 | 9.6 | - | 5.6 | - | 0.3 | 19.7 | 10.5 | 11.0 | - | 57.3 | 31.6 | 56.3 |

## E Other Attempts for Image Generation

Figures 9 and Table 13 illustrate the alternative approaches we explored before arriving at the conclusion that synthetic scene-level images offer the optimal solution for open vocabulary instance segmentation. These methods primarily relied on per-mask rendering, i.e. generating multiple 2D images for each mask.

**Attempts I, II:** Inspired by the success of LAR Bakr et al. (2022), which uses synthetic images of objects to assist in the 3D visual grounding task, we first explore a similar approach. This involves

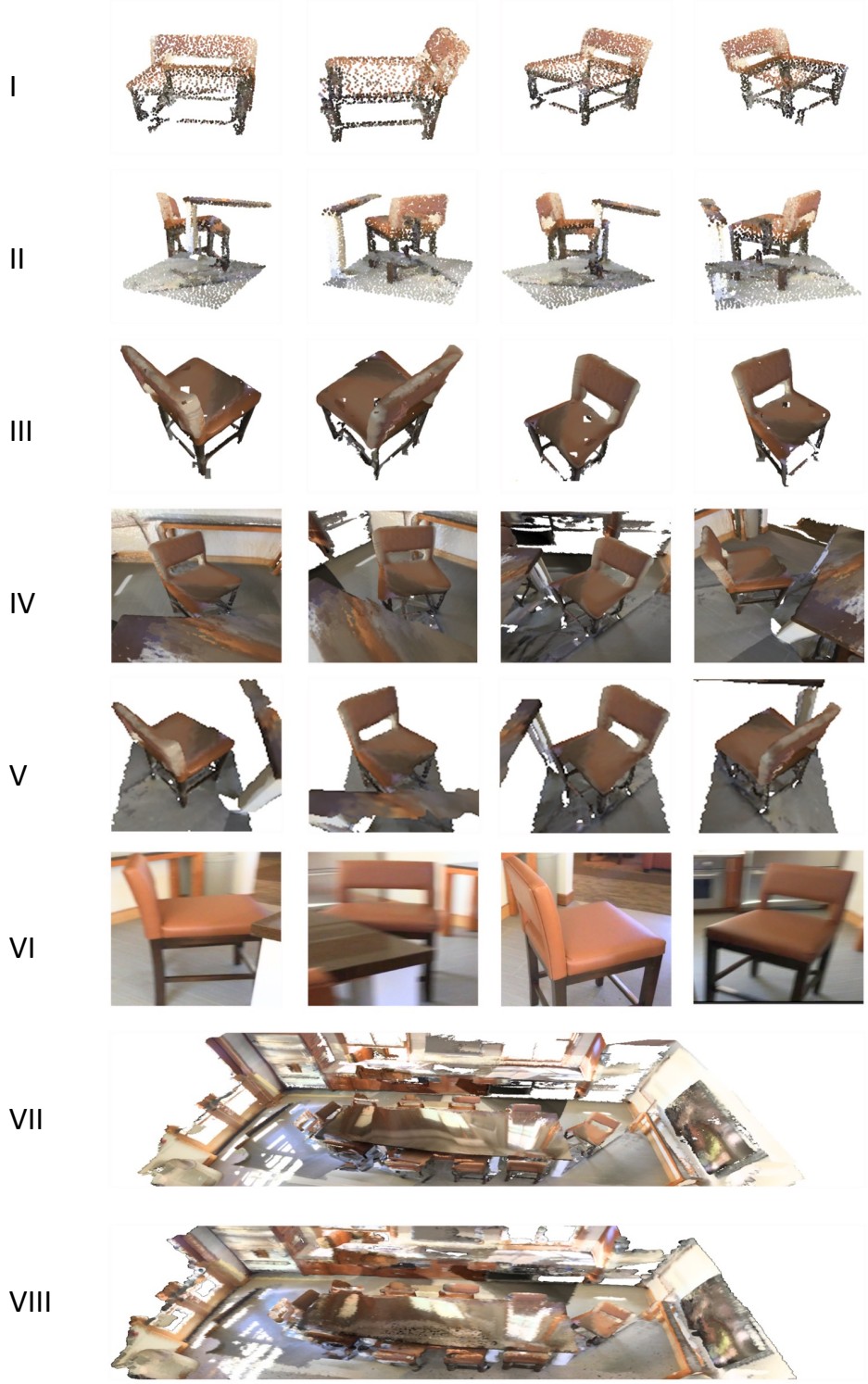

Figure 9: **Visualization of Attempts Made to Generate 2D Images from 3D.** I: LAR-point projection; II: LAR-point-bg-project; III: Mesh rendering; IV Mesh-in-scene Rendering; V: Mesh-bg-Rendering; VI: Cropped from Original 2D images; VII: Scene Level Rendering from Mesh; VIII: Scene Level Rendering from Point. Performance can be found in Table 13.

Table 13: **Evolution of *Snap* and *Lookup* Module.** The corresponding image visualization is shown in Figure 9. Scene-level rendering not only requires fewer images but also achieves superb results when compared to other pre-mask levels of rendering. *: The image sizes of VI are adjusted to fit the size of the mask area on the original images.

| Idx | Methods | Job intensity | Imgs needed | Original 2D | Img size | 2D backbone | AP50 | AP25 |
|---|---|---|---|---|---|---|---|---|
| I | LAR-point projection | per mask | 250 | ✗ | $128^2$ | CLIP | 5.3 | 8.6 |
| II | LAR-point-bg-projection | per mask | 250 | ✗ | $128^2$ | CLIP | 6.3 | 10.5 |
| III | mesh-rendering | per mask | 250 | ✗ | $128^2$ | CLIP | 6.8 | 7.2 |
| IV | mesh-scene-rendering | per mask | 250 | ✗ | $128^2$ | CLIP | 6.7 | 7.3 |
| V | mesh-bg-rendering | per mask | 250 | ✗ | $128^2$ | CLIP | 4.3 | 5.3 |
| VI | crop-original2d | per mask | 250 | ✓ | $-$* | CLIP | 24.3 | 29.6 |
| VII | scene-mesh-rendering | per scene | 8 | ✗ | $1000^2$ | ODISE | 18.8 | 29.8 |
| VIII | scene-mesh-rendering | per scene | 8 | ✗ | $1000^2$ | ODISE | 28.7 | 38.9 |
| IX | scene-point-rendering | per scene | 8 | ✗ | $1000^2$ | ODISE | 21.5 | 33.6 |

positioning the camera around the object and projecting point clouds to generate multi-view images for each mask. However, these approaches yielded unsatisfactory performance when integrated with the CLIP model, even with their background being projected. This is mainly due to the fact that many masks are too broken, and the projected images are difficult to recognize even for humans.

**Attempts III, IV, V:** We redirected our attention to the mesh model. Although meshes are not always available for all 3D point clouds, we decided to investigate whether the mesh model could enhance the quality of rendered images, thereby making them more recognizable with 2D models. However, the outcomes of pre-mask rendering III, IV, V still encountered challenges in achieving reasonable performance, not to mention the considerable rendering time they demanded. The problem still boils down to the quality of the point clouds themselves. For masks with very clear and complete point clouds, such as the chair presented in Figures 9, these approaches can produce reasonable results. However, most masks are very difficult to recognize even in the mesh model. Even for human beings, a substantial amount of contextual information is required to understand those broken, distorted, and sparse mask instances.

**Attempt VI:** We then start to experiment with using original images and cropping out masks in the images for evaluation VI. We believe this offers the best quality of images, therefore making them most likely to be recognizable with 2D models. We use *Occlusion Reports* methods to select the top $K$ views from all frames and crop out mask pixels with an enlarged bounding box. This approach did achieve notable performance, primarily due to the high quality of 2D images. **However, we ultimately abandoned this approach due to concerns about its applicability in general scenarios for two reasons:**

1. Such an approach requires well-aligned 2D images in both the training and inference stages. Our argument is that if well-aligned 2D images are readily available and can be seamlessly linked with 3D data, meaning they have pose, intrinsic, and depth information, it is much easier to approach open-world tasks from a 2D perspective. The open-world understanding of 3D can be achieved by conducting open-world detection in 2D images and projecting the results into 3D point clouds using available camera models and depth maps. Similar approaches have been shown to be feasible in SAM3D Yang et al. (2023b). Therefore, the motivation for such an approach is questionable.

2. In many instances, 2D images occupy a significant portion of storage space, and it would be more practical to not rely solely on 2D images. For example, in ScanNetv2, the point cloud/mesh file of a scene occupies around 3MB, while the entire 2D image collection consumes roughly 3GB. Additionally, in many cases, 2D images are not available, as discussed in the introduction section.

**Attempt VII, VIII, XI:** Turning our attention to scene-level rendering, our model demonstrated a significant enhancement in performance (VII) with a notable increase of 12.0% on AP50. This was because, by observing all broken instances from a distance and incorporating a large amount of contextual information, objects became clear and recognizable. Switching the 2D backbone from Grounding-DINO to ODISE offers further improvement (+8.9%), as ODISE proves to be more

robust for diverse input queries. While rendering by the mesh model contributed to improved clarity (VIII) in images, leading to enhanced results, rendering from the point cloud made the approach more applicable (XI), and the performance remained decent when compared to other state-of-the-art methods. This is why we rendered from the point cloud in most of the datasets we tested.

## F    LIMITATIONS AND FUTURE WORK

OpenIns3D is a novel framework that achieves remarkable performance in open-vocabulary instance segmentation, surpassing many existing methods. However, there are some limitations of OpenIns3D that need further investigation in future studies.

- Reliance on Ground Truth Instance Masks: Similar to SAM Kirillov et al. (2023), OpenIns3D still relies on ground truth mask supervision. While it does prove to have the capability to generalize masks that have never been seen before, the performance on mask proposal at scale still has a large room for improvement. Learning from the success of SAM, a query-based transformer decoder backbone could yield impressive results when trained with a large-scale dataset. There is a line of research dedicated to learning 3D class-agnostic masks, exemplified by approaches like UnScene3D Rozenberszki et al. (2023) and SAM3D Yang et al. (2023b). These approaches could serve as either alternatives to MPM or as a means of generating class-agnostic labels at scale for MPM to be trained on. Exploring this path further could be an interesting avenue for future research.

- Limited Performance in Semantic Segmentation: OpenIns3D heavily relies on filtering to refine the mask proposals, discarding masks with low quality directly. While this approach benefits instance segmentation by reducing false positive instances, it may limit its performance in semantic segmentation. We have also calculated the semantic segmentation results of OpenIns3D on four categories, as reported by OpenScene Peng et al. (2023), as shown in Table 14. Our method still exhibits a gap compared to OpenScene in terms of semantic segmentation.

- Small Object Performance: As shown in Table 8, the performance of OpenIns3D is ultimately closely linked to the quality of the point cloud itself. Masks that are very small or made of sparse point clouds would be difficult to recognize in the rendered images, as they either occupy a small portion of the image pixels or are too fragmented to be detected by the 2D models.

Nonetheless, while most researchers in the community are focused on aligning point and image features for open-world capabilities in the 3D domain, we aim to propose a simple, flexible, and powerful framework that requires no 2D input but can still achieve impressive results. OpenIns3D can easily evolve with the rapid development of 2D open-world models. In this era of rapid evolution of foundation models, we believe this attribute makes OpenIns3D powerful in many settings.

Table 14: **Comparison with OpenScene and other Frameworks on Semantic Segmentation.** Our framework prioritises mask quality and suffers overall semantic segmentation results.

| Semantic Seg. | mIoU | | | | | mAcc | | | | |
|---|---|---|---|---|---|---|---|---|---|---|
| Methods | Bookshelf | Desk | Sofa | Toilet | Mean | Bookshelf | Desk | Sofa | Toilet | Mean |
| 3DGenZ (Michele et al., 2021) | 6.3 | 3.3 | 13.1 | 8.1 | 7.7 | 13.4 | 5.9 | 5.9 | 26.3 | 12.9 |
| MSeg Voting (Lambert et al., 2021) | 47.8 | 40.3 | 56.5 | 68.8 | 53.3 | 50.1 | 67.7 | 67.7 | 81.0 | 66.6 |
| OpenScene-LSeg (Peng et al., 2023) | **67.1** | **46.4** | 60.2 | **77.5** | **62.8** | **85.5** | **69.5** | 69.5 | 90.0 | 78.6 |
| OpenScene-OpenSeg (Peng et al., 2023) | 64.1 | 27.4 | 49.6 | 63.7 | 51.2 | 73.7 | 73.4 | 73.4 | 95.3 | 79.0 |
| OpenIns3D | 54.8 | 16.7 | **61.6** | 50.6 | 45.9 | 59.0 | 32.3 | **76.7** | 79.8 | 61.9 |

## G    VISUALIZATION

**Mask Proposal**    Figure 10 and 11 present a qualitative evaluation of the mask proposal module. The learned mask proposals exhibit great similarity to the ground truth masks, often capturing additional unlabeled masks. This demonstrates the effectiveness of our class-label-free learning scheme in producing high-quality class-agnostic mask proposals. Moreover, through the application of *Mask*

*Scoring* and *Mask Filtering* techniques, we are able to connect fragmented or fragile masks, resulting in a substantial improvement in mask quality. These advancements provide a strong foundation for the Snap and Lookup understanding scheme.

**Snap visualization**  Figure 12, 13 and 14 demonstrate the capability of the Snap module. With the proposed pose and intrinsic optimization scheme, the Snap module is capable of generating decent-quality images from point clouds, regardless of whether the dataset is indoor or outdoor.

**Lookup Results Visualization**  The Lookup module effectively links 2D results with 3D. Here, we present visualizations of its outcomes from all three datasets (Figure 15, 16, 17). OpenIns3D is capable of capturing the most interesting objects in the scene without relying on any corresponding 2D images.

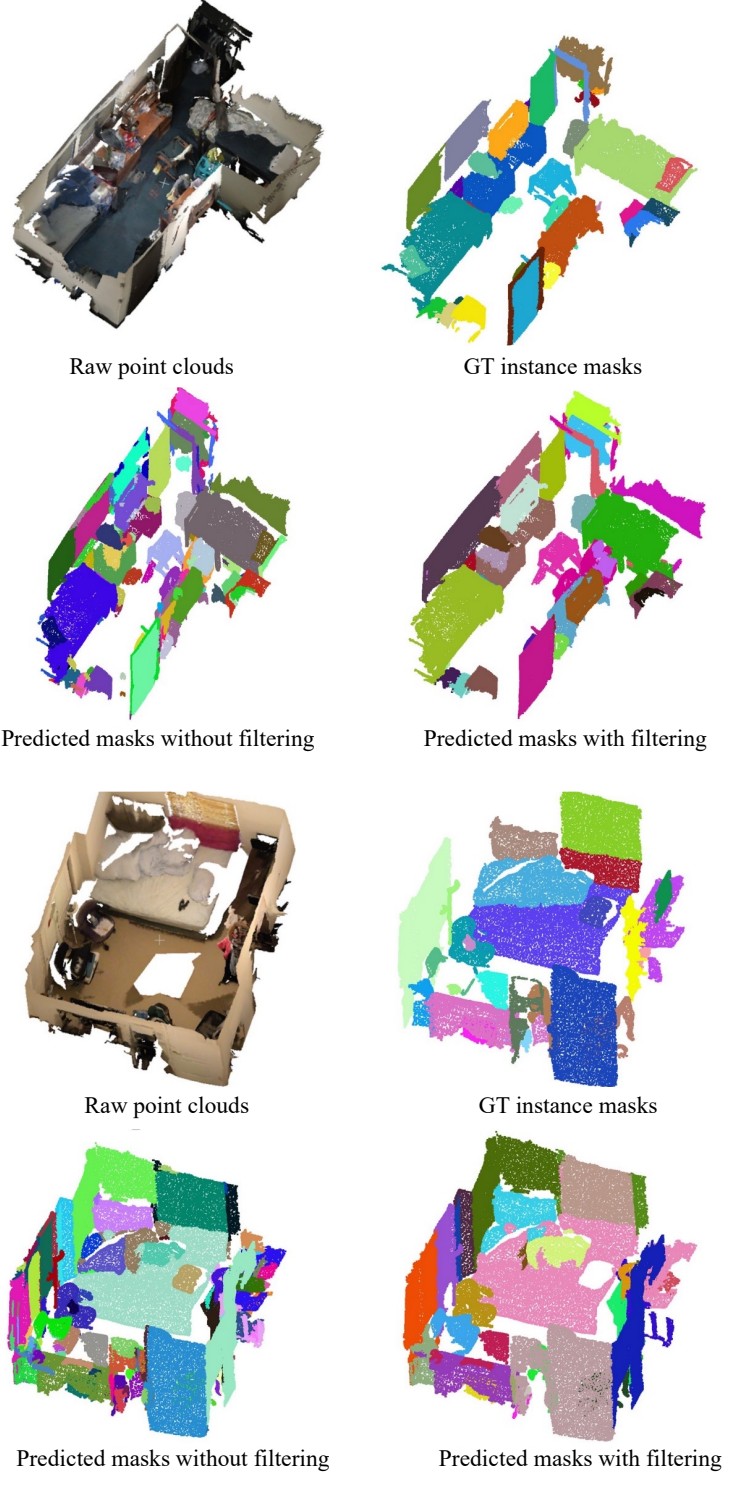

Figure 10: **Qualitative Evaluation of the Mask Proposals.** Our class-label-free approach produces high-quality masks that closely resemble the ground truth. Additionally, the incorporation of *Mask Scoring* and *Mask Filtering* further enhances the overall quality of the masks. Quantitative evaluation is shown in Table 5.

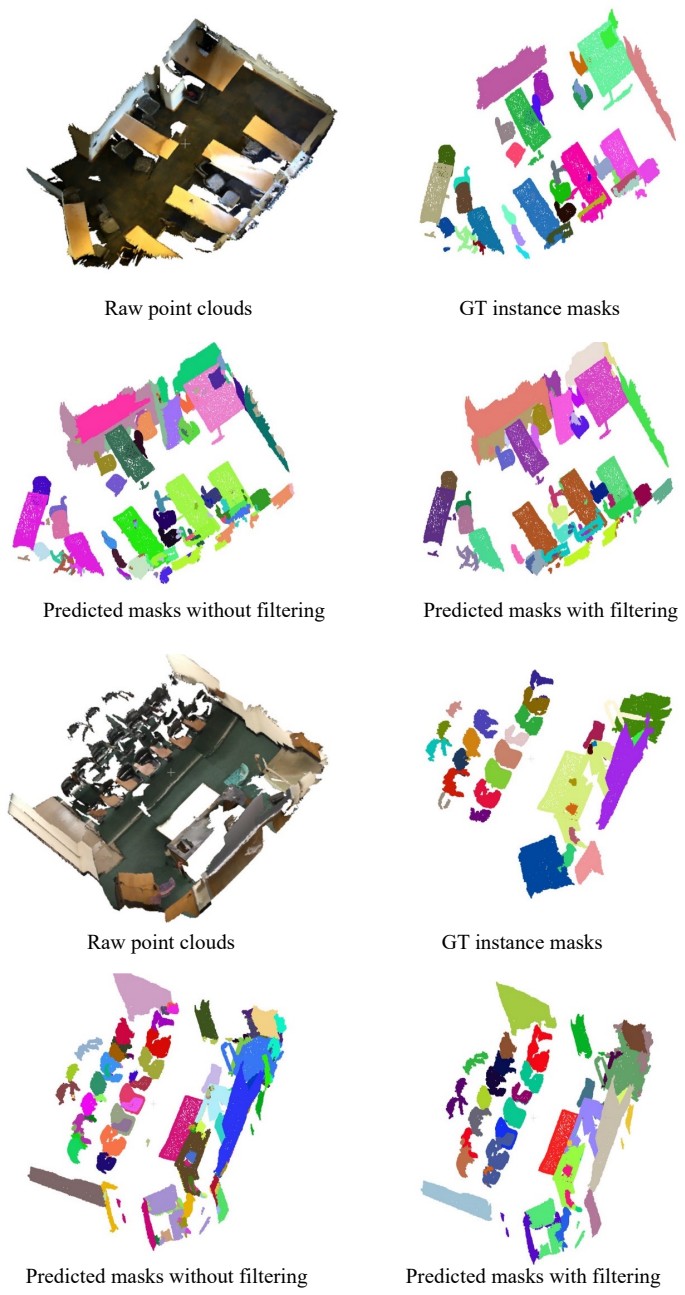

Figure 11: **Qualitative Evaluation of the Mask Proposals.** Our class-label-free approach produces high-quality masks that closely resemble the ground truth. Additionally, the incorporation of *Mask Scoring* and *Mask Filtering* further enhances the overall quality of the masks. Quantitative evaluation is shown in Table 5.

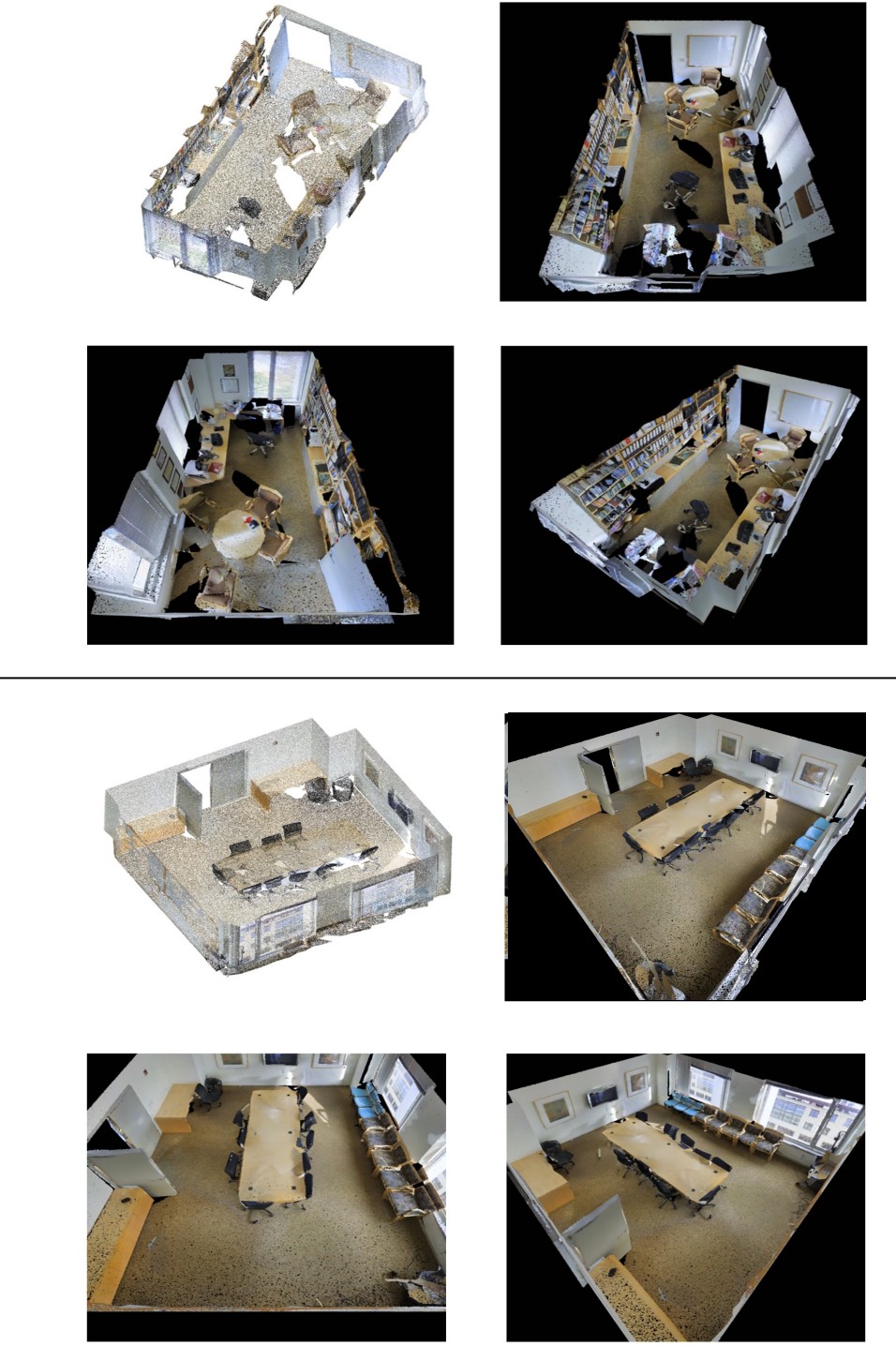

Figure 12: **Synthetic Scene-level Images of S3DIS Generated by *Snap*.** The first image is the original spare point cloud, and the following three images are outcomes of the *Snap* module.

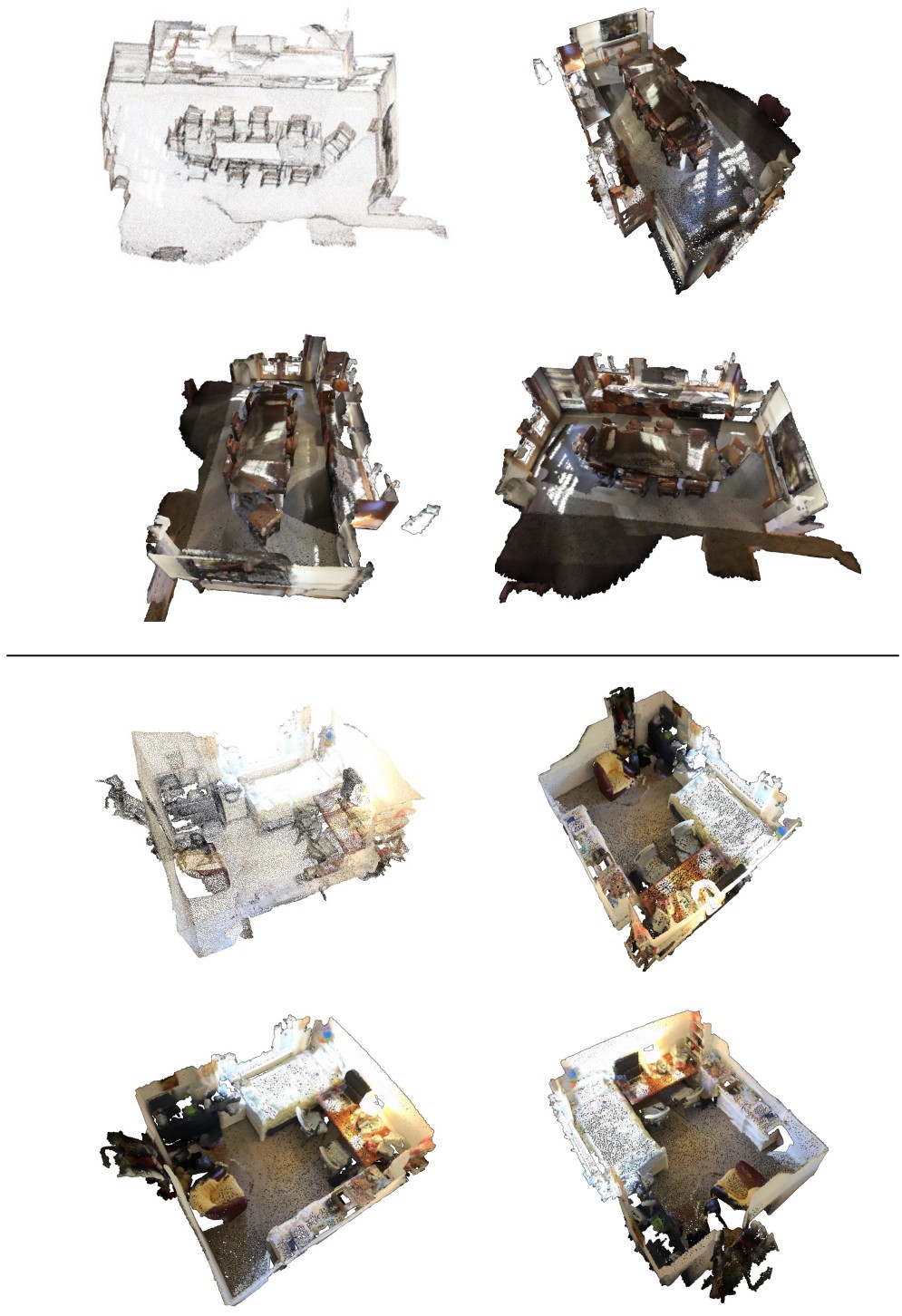

Figure 13: **Synthetic Scene-level Images of ScanNetv2 Generated by *Snap*.** The first image is the original spare point cloud, and the following three images are outcomes of the *Snap* module.

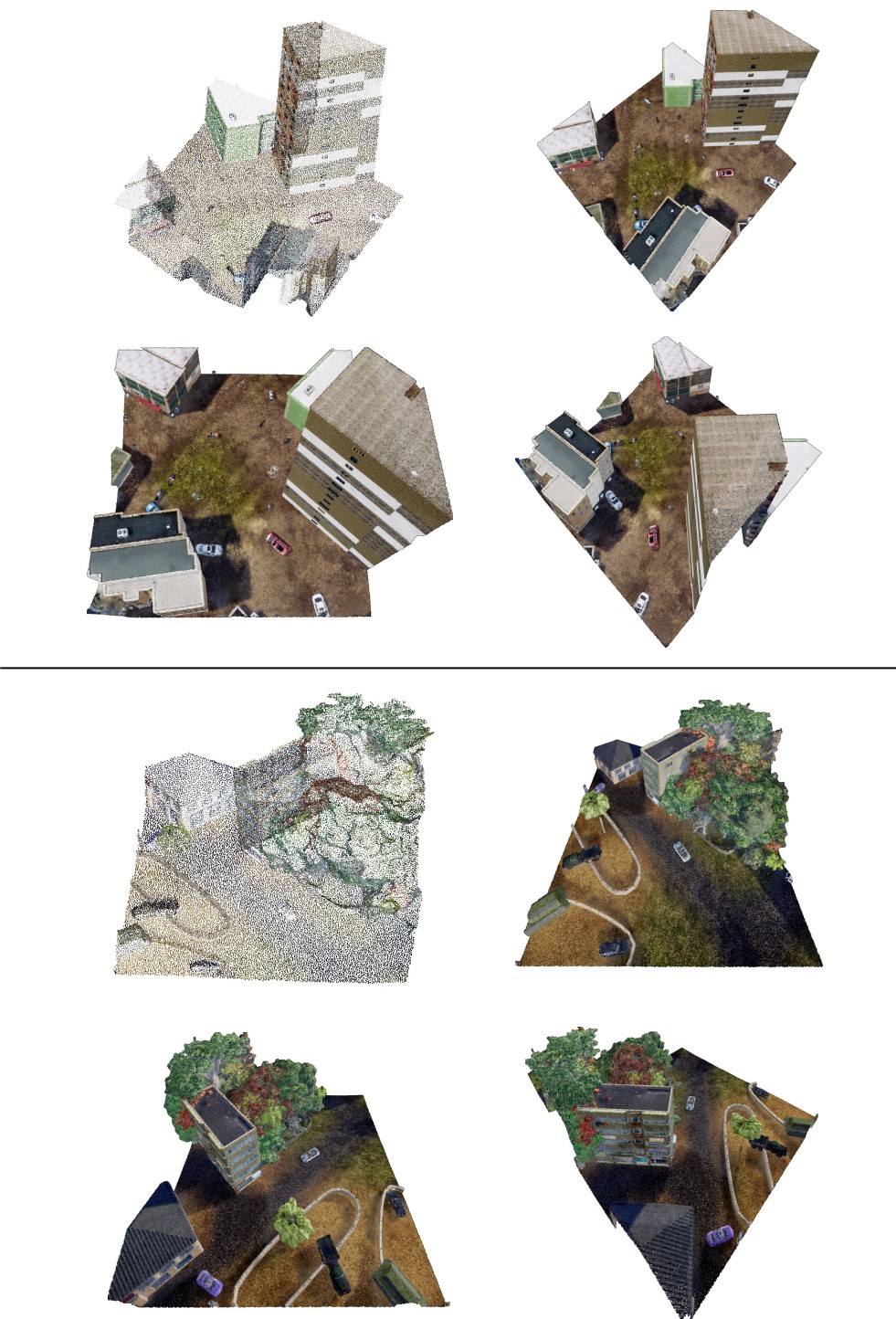

Figure 14: **Synthetic Scene-level Images of STPLS3D Generated by *Snap*.** The first image is the original spare point cloud, and the following three images are outcomes of the *Snap* module.

Input Queries: *'ceiling', 'floor', 'wall', 'beam', 'column', 'window', 'door', 'table', 'chair', 'sofa', 'bookcase', 'board'*

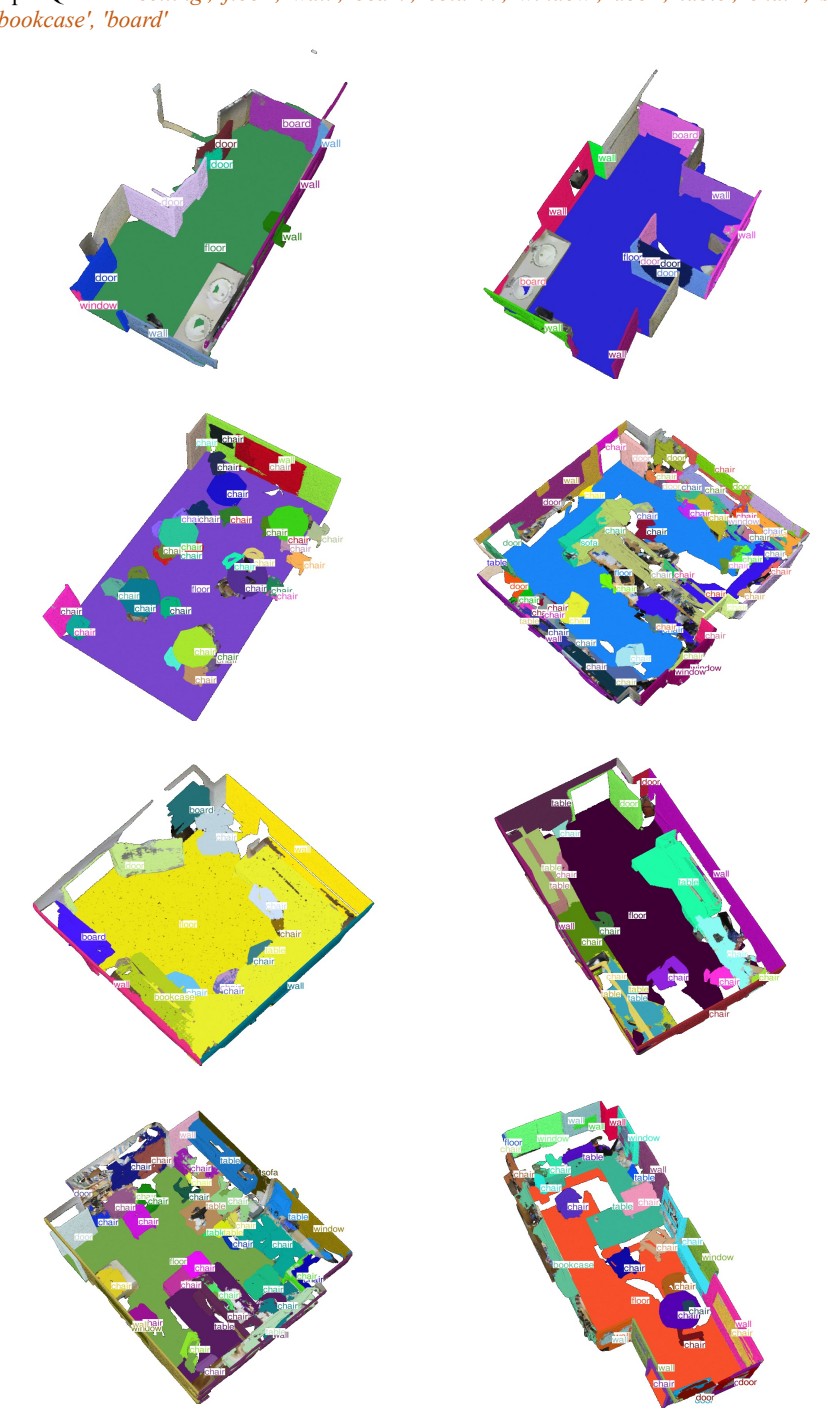

Figure 15: **Open-vocabulary Instance Segmentation Results of S3DIS by OpenIns3D (ODISE).** Instance and class labels are presented in the same color.

Input Queries: *'cabinet', 'bed', 'chair', 'sofa', 'table', 'door', 'window', 'bookshelf', 'picture', 'counter', 'desk', 'curtain', 'refrigerator', 'shower curtain', 'toilet', 'sink', 'bathtub'*

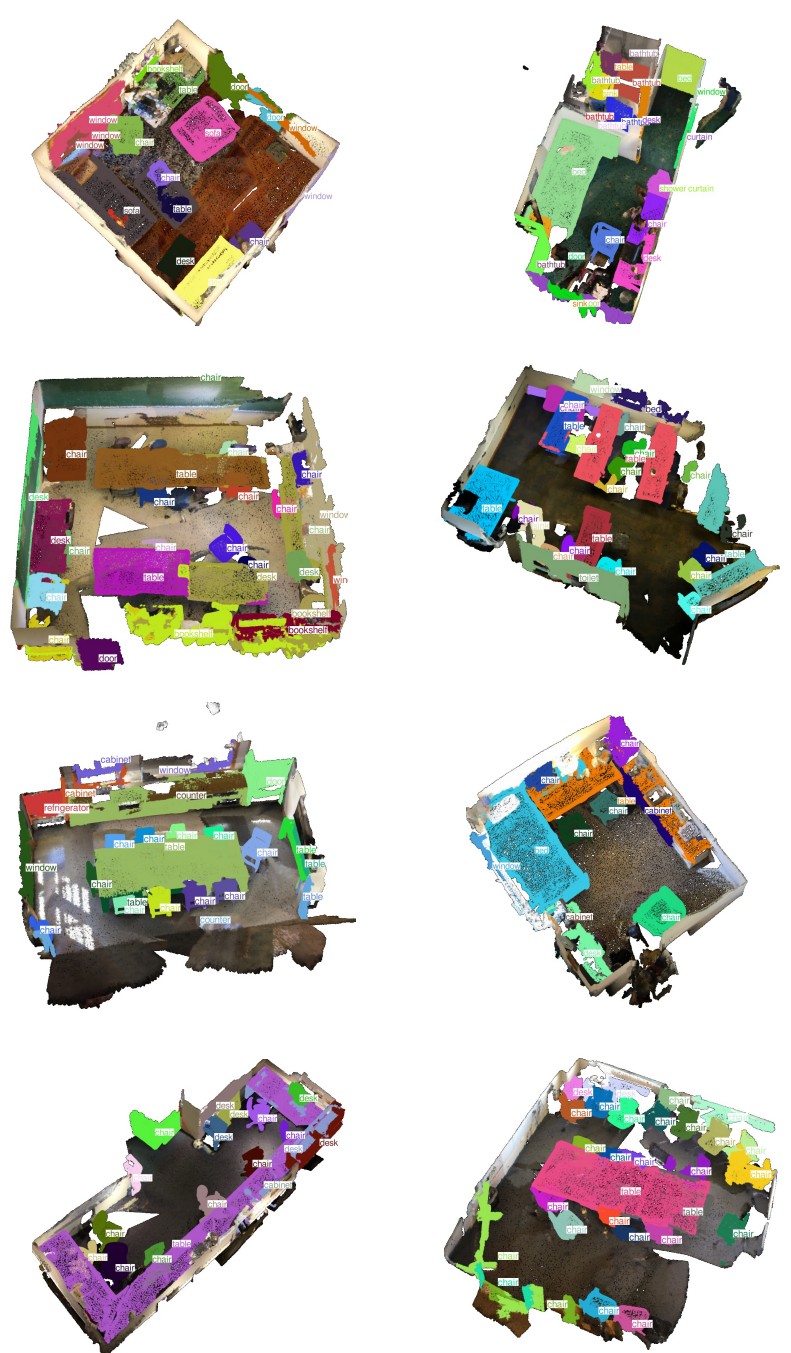

Figure 16: **Open-vocabulary Instance Segmentation Results of ScanNetv2 by OpenIns3D (ODISE).** Instance and class labels are presented in the same color.

Input Queries: *'building', 'vegetation', 'vehicle', 'truck', 'Aircraft', 'military vehicle', 'bike', 'motorcycle', 'light pole', 'street sign', 'clutter', 'fence'*

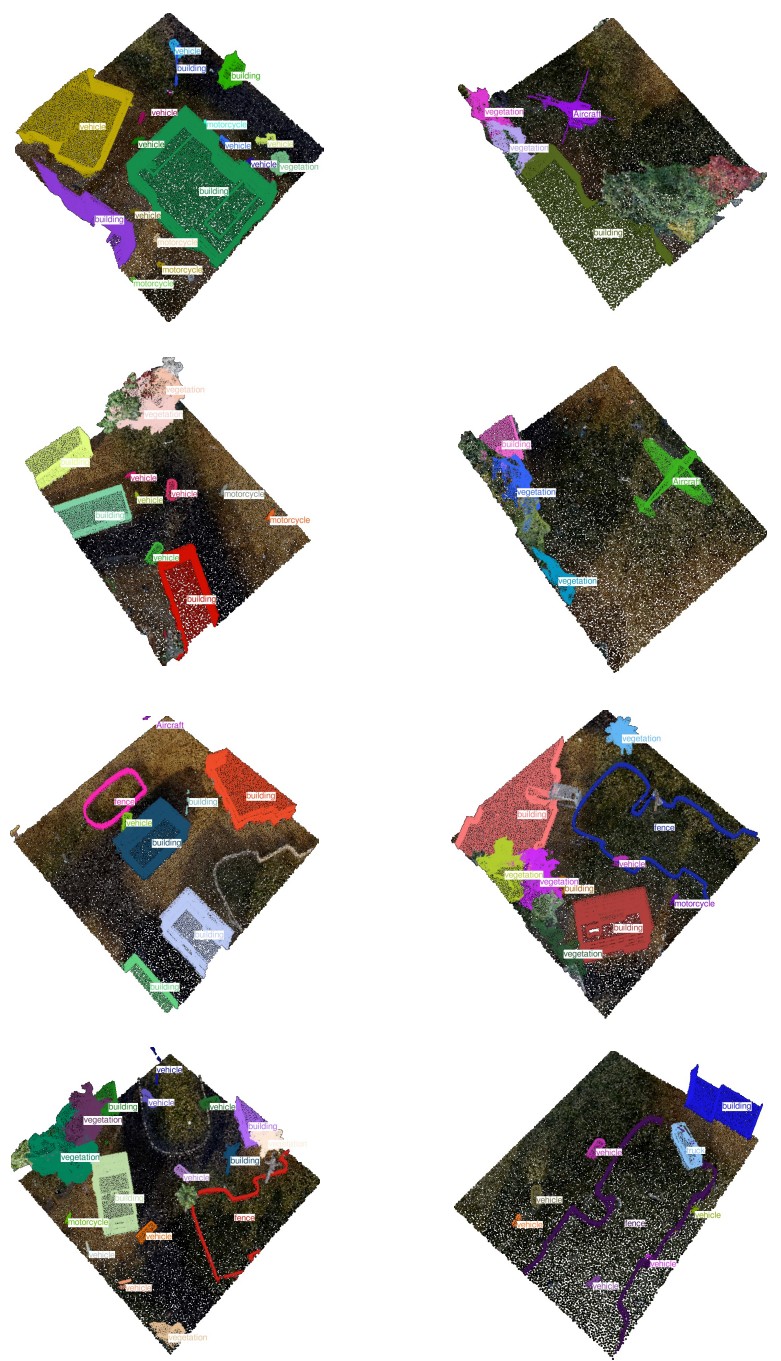

Figure 17: **Open-vocabulary Instance Segmentation Results of STPLS3D by OpenIns3D (ODISE).** Instance and class labels are presented in the same color.

