# OpenReview forum: "OpenIns3D: Snap and Lookup for 3D Open-vocabulary Instance Segmentation"
_ICLR.cc/2024/Conference — Submitted to ICLR 2024_

### Official Review · Reviewer_3Uq7 · 2023-10-25

**Soundness:** 3 good
**Presentation:** 3 good
**Contribution:** 3 good
**Rating:** 6
**Confidence:** 5

**Summary:**

This paper focuses on open-vocabulary 3D instance segmentation. They introduce a new pipeline, namely, OpenIns3D, which requires no 2D image inputs, for 3D open-vocabulary scene understanding at the instance level. The OpenIns3D framework employs a “Mask Snap-Lookup” scheme. The “Mask” module learns class-agnostic mask proposals in 3D point clouds. The “Snap” module generates synthetic scene-level images at multiple scales and leverages 2D vision language models to extract interesting objects. The “Lookup” module searches through the outcomes of “Snap” with the help of Mask2Pixel maps, which contain the precise correspondence between 3D masks and synthetic images, to assign category names to the proposed masks. This 2D input-free and flexible approach achieves state-of-the-art results on a wide range of indoor and outdoor datasets by a large margin.

**Strengths:**

- Open-vocabulary instance segmentation is an important research topic in 3D scene understanding, this paper proposed a novel method to tackle this problem.

- They propose a novel Mask-Snap-Lookup scheme, which distills knowledge from 2D foundation models to the 3D masked point clouds.

- Their method outperforms the existing baseline approaches on several benchmarks, which demonstrates the effectiveness of their proposed method.

- The paper writing is clear and easy to follow.

**Weaknesses:**

- The mask proposal module is adapted from an existing instance segmentation model Mask3D. Although the authors remove the class-specific information, essentially Mask3D is trained with a close set of categories. Hence the mask proposal module is not class-agnostic and the whole system is not an open-vocabulary system, as this system cannot handle the irregular point cloud clusters that don't belong to the indoor object classes. To evaluate this, I suggest the authors adapt their system to outdoor driving scenarios such as nuScenes, to see whether their approach can generate mask proposals of objects on a road such as traffic cones and barriers.

- The authors need to compare with CLIP^2 [1] in Table 1, which can also generate open-vocabulary instance segmentation results.

[1] Zeng et al. CLIP$^2$: Contrastive Language-Image-Point Pretraining from Real-World Point Cloud Data. CVPR 2023.

**Questions:**

Please refer to the weaknesses.

---

> ### Author Response · Authors · 2023-11-19
>
> Dear reviewer,
>
> Thank you so much for recognizing that 3D open-world instance segmentation is an important topic, as well as acknowledging the novelty and efficiency of our "Mask-Snap-Lookup" framework. We hereby address your other comments as below:
>
> 1. Generalisation of Mask proposal Module
>
> It is widely acknowledged that a substantial domain gap exists between indoor and outdoor scenes in 3D. This is why many attempts at 3D Foundation Models ([1],[2],[3])  and Unified 3D representation [4][5] models still require training separate weights for indoor and outdoor scenes.
>
> It is true that generalizing the indoor scene trained Mask Proposal Module (MPM) to the outdoor scene would be a remaining challenge. However, we believe that within the same scene space, i.e., indoor or outdoor, the MPM possesses a decent level of generalization capability. To substantiate this claim, we conducted zero-shot experiments with OpenIns3D, trained on ScanNet, across four diverse types of indoor datasets. The visual results are presented in the following anonymous repository: https://anonymous.4open.science/r/rebuttal_image-6CBE/README.md.
>
> 2. Comparison with CLIP^2
>
> We thank the review for bringing our attention to this good work. Indeed, CLIP$^2$ can also generate instance-level results with the help of aligned 2D images. The results are rather good on ScanNetV2. We will update the paper and add this to the literature review.
>
> Integrating the results into Table 1 might be challenging, though, as CLIP$^2$, unlike other 3D instance segmentation work, uses the top-1 accuracy metric rather than the Average Precision metrics. It would not be so cohesive to put it in Table 1.
>
> Speaking of Table 1, an interesting insight that might be worth sharing is that, upon slight fine-tuning and testing (better mask proposal and better SNAP images), the performance of OpenIns3D on the novel 17 classes on ScanNet v2 can be massively improved compared to the original reported results. As shown in the table below. We think this indicates that the "Mask-Snap-Lookup" approach has huge potential that can be explored in the future.
>
> |                      |   AP  |  AP50 |  AP25 |
> |:--------------------:|:-----:|:-----:|:-----:|
> |  Mask3D + PointCLIP  |  3.20 |  4.50 | 14.40 |
> | OpenIns3D (reported) | 16.80 | 28.70 | 38.90 |
> |    OpenIns3D (new)   | **28.25** | **38.44** | **46.30** |
>
> We hope this would be helpful, and please do let me know if you have any other questions about the work.
>
>
> [1] PonderV2: Pave the Way for 3D Foundation Model with A Universal Pre-training Paradigm
>
> [2] Towards Large-scale 3D Representation Learning with Multi-dataset Point Prompt Training
>
> [3] PointContrast: Unsupervised Pre-training for 3D Point Cloud Understanding
>
> [4] Uni3D: Exploring Unified 3D Representation at Scale
>
> [5] Exploring Data-Efficient 3D Scene Understanding with Contrastive Scene Contexts

---

### Official Review · Reviewer_A44f · 2023-11-01

**Soundness:** 3 good
**Presentation:** 3 good
**Contribution:** 3 good
**Rating:** 6
**Confidence:** 3

**Summary:**

This paper proposes a novel pipeline called OpenIns3D, which consists of three core steps: mask, snap, and lookup, eliminating the need for 2D image inputs and enabling 3D open-vocabulary scene understanding at the instance level.  This approach not only requires less rendering time and inference time but also achieves much stronger results.

**Strengths:**

1. The experimental results are impressive. The OpenIns3D achieves superior quantitative results compared with other methods.
2. The idea is interesting. The authors propose a novel framework that can achieve 3D open-vocabulary scene understanding without 2D images.
2. This paper is well-written and maintains a smooth flow. The whole pipeline is easy to understand.

**Weaknesses:**

1. As the method consists of multiple steps, the authors should provide more training details for all steps in the main text or appendix.
2. Will the performance of 2D Open-world Detector influence the performance of the OpenIns3D? The authors seems not to provide experimental results in ablation study.
3. Although the authors propose a 3D open-vocabulary scene understanding without 2D images, this method still needs well-prepared point clouds. It seems that 3D point clouds are also difficult to obtain in real world.

**Questions:**

1. Input queries shown in experiments are a few words. The authors should provide additional experimental results on processing real whole sentences.
2. The authors should provide visual comparisons with SOTA.

---

> ### Author Response · Authors · 2023-11-17
>
> Dear reviewer,
>
> We would like to start by thanking you for your feedback and your support of our work. It is great to see that you found the idea interesting. We reply your comments as below.
>
> ## More details in the paper
> As per our response to reviewer (rePE), we will include more information about Mask Scoring Module, as well as the Snap module in the revision (will be added later). More details about these modules can be found in the reply to the reviewer (rePE).
>
> ## Will the 2D performance influence the performance of 3D?
>
> Indeed. This is because we completely outsource the detection tasks to 2D model. We provided a small piece of comparison results in Table 2, but here we offer a direct comparison of OpenIns3D using two different backbones, as shown in the table below.
>
> |Methods|Rendering|2Dbackbone|AP50|AP25|
> |-|-|:-:|:-:|:-:|
> |Mask-P-CLIP|Mask|CLIP|4.5|14.4|
> |OpenIns3D|Scene-level|Grounding-DINO|18.8(+14.4)|29.8(+15.4)|
> |OpenIns3D|Scene-level|ODISE|28.7|38.9|
>
> An important note here is that once the rendering is switched to the scene level, the general performance improves significantly compared to that of the object/mask-level rendering, regarding less the 2D model used.
>
> A further note is that relying on the 2D detector, on the other hand, is also a good strategy, as their evolving speed is generally faster. This ensures that OpenIns3D can also evolve with the 2D detector. This design also offers more special abilities, such as understanding complex sentences when the 2D detector is integrated with the LLM. We demonstrated this in the paper in Figure 1 and 6.
>
> ##	Does the point cloud need to be well-prepared?
>
> To a certain extent, yes. At least RGB information is required, which is very common for all point clouds.
> To demonstrate the performance of OpenIns3D on various point cloud datasets, we conducted quick zero-shot experiments on four diverse types of datasets. We showcase the results in the following **anonymous repository**:
>
> **https://anonymous.4open.science/r/rebuttal_image-6CBE/README.md**
>
> Please feel free to check it out.
>
> In conclusion, since OpenIns3D does not rely on aligned images, it can be deployed and used for 3D data rapidly. It also exhibits a relatively good generalisation capability on various datasets.
>
> ## Questions:
> - Can the model work on long sentence queries? Yes, and this might also be a compelling advantage of OpenIns3D. While other CLIP-based models would perform reasonably on singular vocabulary or phrases, OpenIns3D, equipped with the right 2D detector model, can understand long sentences and carry out segmentation tasks.
>
> - Due to time constraints in the rebuttal period, we might not be able to include a comparison with other state-of-the-art models, but we will certainly include this as future work.
>
> Thanks again for your input, and please let us know if you have any further questions regarding the paper.

---

> > ### Comment · Reviewer_A44f · 2023-11-22
> > **Official comments by Reviewer A44f**
> >
> > Thanks for the author’s detailed responses. Based on other reviewers’ feedback and the authors’ responses, I will keep my rate.

---

> > > ### Author Response · Authors · 2023-11-23
> > >
> > > Dear Reviewer A44f,
> > >
> > > We would like to thank you for your insightful feedback and support of our work. We believe upon the realise of our code, OpenIns3D will be very valuable for the domain of 3D open world scene understanding.
> > >
> > > Best,
> > >
> > > OpenIns3D team

---

### Official Review · Reviewer_wuwY · 2023-11-01

**Soundness:** 2 fair
**Presentation:** 2 fair
**Contribution:** 2 fair
**Rating:** 3
**Confidence:** 4

**Summary:**

The paper proposes a method for open-vocabulary instance segmentation. It consists of three main steps, first around a 3D point cloud of a scene, several virtual cameras are placed (all pointed inwards) to record several synthetic images of the scene (snap). Some 2D open-vocabulary segmentation or detection approach is then applied to these images to find the sought after objects. In parallel, a class agnostic variant of Mask3D is used to extract object proposals (mask). Finally, the obtained class agnostic masks are matched to the obtained open vocabulary instances to assign them to a class (lookup). The key difference to previous methods is that this approach does not require aligned RGB images to be present in the data, but rather relies on synthetically created images to be fed into a 2D vision-language model. Compared to previous open vocabulary methods evaluated on ScanNetV2, S3DIS, and STPLS3D, the method achieves better performances.

**Strengths:**

- The writing of the paper is easy to follow.
- The paper tackles the interesting task of OV point cloud instance segmentation.
- The scores compared to some baselines look promising, even without the use of 2D images.

**Weaknesses:**

- At its core, the method is built on top of a somewhat flawed assumption. How can we obtain RGB point clouds, without actually having aligned RGB images? Of course there might be LiDAR point clouds without aligned RGB images, but at that point we can also not create synthetic RGB images from an uncolored point cloud, to feed into a 2D model expecting RGB images. While I still see some potential benefit, like being able to render novel images that are more focused on certain objects or better suited for the downstream model, this aspect is not explored here. I can't really imagine another case, where we have RGB point clouds, but for some reason we had to delete each and every underlying RGB frame used for colorization. As such I don't see the true merit of this approach.

- Ignoring the above issue, the "mask" module is not novel, it's very similar to the one used in OpenMask3D with some mask filtering from SAM, the "lookup" module is fairly simple matching and not super interesting as far as I can tell. Leaving the "snap" module to be the core novelty of this approach. The way it is described initially: "Multiple synthetic scene-level images are generated with calibrated and optimized camera
poses and intrinsic parameters." gives the impression that the camera poses and intrinsics might somehow be optimized on a scene by scene basis, to create the best possible synthetic images for a given scene, resulting in the best scores. However, I understand this module simply places a predefined number of cameras around the scene, looking at the center and optimizing the focal length according to some heuristics. The slightly more involved "local enforced lookup" is mostly explained in the supplementary, raising the question what the real novelty of this paper is supposed to be? Finally, the rendered images are still just points splatted to the camera? According to "Challenge 4" there is a domain gap between projected and natural images, which I agree with, but as far as I can tell, the paper does nothing to truly bridge this gap, apart from aiming virtual cameras in a certain way?

- The training of Mask3D for class mask predictions can have a significant effect on how truly "open vocabulary" the downstream method really is. The way all of these methods are presented is that one can easily find objects as long as we can describe them. This method relies on Mask3D being able to create class agnostic masks for the objects though. To thus evaluate how well it generalized to novel classes, Mask3D should not be trained on these classes. Now I'm assuming that Mask3D was trained to exactly create proposals for the classes also evaluated on, which makes the whole evaluation questionable. The same weakness is actually also a problem in the evaluation on OpenMask3D. While the paper does show some qualitative results in the teaser figure, this does not really prove the overall generalization. To properly show this, it could have been a great opportunity to simply evaluate on ScanNet200, given this should not require any novel training.

- In general the lack of evaluation on a larger set of classes such as ScanNet200 is a clear weakness in my opinion. In fact, if you can provide such results in the rebuttal, without retraining (which should thus be fast), I would be happy to change my opinion. This would also allow a direct comparison with OpenMask3D. And yes, while one might claim that this was not published prior to the ICLR deadline, the paper clearly acknowledges OpenMask3D exists by citing it and even contains a figure about it. As such a simple comparison would be more than fair.

- An actual ablation about how real images compare to synthetic images for this would have been very valuable. An interesting setup could have been to use a set of real images from the dataset and compare it to rendered synthetic images using the same camera pose and intrinsics.

- Certain parts of the text make it seem like it's a bad thing to use a pre-trained network such as 3DETR or DETR, clearly stating that MPM is trained from scratch (start of chapter 5). This feels a bit hypocritical given that this whole method relies on pretrained 2D VL models trained on huge amounts of data. In general I must sadly say the text seems to contain a lot of "fluff" for me, introducing new fancy names for all kinds of stuff.

**Questions:**

- I think that the references to Table 5 and 6 in the text are swapped.

- The introduction states "and point clouds generated by photogrammetry frequently lack depth maps", how is this relevant here? And is a point projected to an image not a sparse depth map?

- As suggested above, an actual evaluation on ScanNet200 would make a major difference.

---

> ### Author Response · Authors · 2023-11-11
>
> We thank the reviewer for the insight feedback, the following are our response:
>
> Weakness session:
>
> ## Item 1:
>
> The reviewer argues that since RGB point clouds are derived from RGB images (a point we acknowledge), developing a method that intentionally excludes the use of these 2D images for 3D scene understanding may not be considered meaningful (a viewpoint we strongly disagree with). Following this line of reasoning, it implies that the majority of point cloud processing methods, ranging from PointNet to PointTransformer, SpareNet, and Mask3D, which solely utilize 3D RGB point clouds as input without incorporating 2D images, do not offer a clear advantage over methods that demands both modalities as input. This argument is not valid logically.
>
> When OpenIns3D will be useful:
> Imagine you've been given only a RGB 3D point cloud/mesh and need to conduct 3D open-world instance understanding. OpenIns3D remains easily applicable, while other methods may suffer.
> More use cases, are explained in the introduction session, here we provide a condensed summary.
> - LiDAR point cloud often exists without matched 2D images.
> Example: Semantic3D, Tornoto3D, Paris-carla-3d, etc.
> - Photogrammetry point cloud often exists without depth maps. [without depth maps it is hard to build point-pixel correspondence]
> Examples: Sensaturban, Campus3d, STPLS3D, etc.
> - Point cloud generated from the registration of multiple scans or converted from 3D simulations/CAD models typically lacks 2D images.
> Example: SynthCity, etc.
>
> (Note: Example datasets are publicly available dataset under these described circumstances.)
>
> In real Application, we may loss of pose, intrinsic, and depth information in the transferring process; we may remove of 2D images to reduce storage requirements. Example: ScanNetv2, 3D scan (3MB), image package (3GB). Clearly saving a 3MB mesh is easier than saving entire 3GB images.
>
> ## Item 2:
>
> Before addressing any questions related to this item, we wish to emphasize our belief that good and meaningful research lies not in proposing sophisticated methods but in suggesting simple yet effective approaches that yield sustainable gains for the tasks and new insights for the community.
>
> - Substantial Gains:
> OpenIns3D has proposed a 2D input-free framework, which is not only more flexible to use but has also achieved a wide range of state-of-the-art results in indoor and outdoor, instance segmentation and object detection tasks. This success is evident even when compared with methods using original 2D images.
>
> - New Insights: One of the main insights of "Mask-Snap-Lookup" is that scene-level rendering scheme is beneficial for 3D open-world understanding. Because scene-level images enable all masks to be rendered and comprehended at once, massively reducing inference time while providing robust results. Consider having 100 masks in a scene; other approaches that handle each mask one by one need to render/crop 200-400 images, while scene-level rendering approach only requires 8-16 images, which provide a significant benefits in overall runtime. Meanwhile, this new pipeline can seamlessly evolve with 2D open world models without retraining, and handle super complex text input when integrated with an LLM-powered 2D detector.
>
> The Mask module indeed leverages the strengths of both Mask3D and SAM, and our scoring and filtering design makes it work really well, providing a solid foundation for further steps.
> As previously explained, the Snap-Lookup module is designed to enable scene-level rendering. The lookup approach is not a simple match. The key idea behind the Lookup module is the construction and use of a mask2pixel map. 2D models work best when the image contains more background context and mask2pixel map allows us to search precisely in a global image for a specific 3D mask. We believe this design would be very useful for many 3D open-world models intending to leverage 2D image results.
>
> Gap with the natural image: Point clouds are rendered into images using ray tracing add proper lighting rather than simple point projection.
>
> As explained in the appendix (Fig 9), when rendering images for each mask individually—whether using PointClip rendering or LAR rendering—these images are challenging for 2D models to comprehend. Transitioning to scene-level rendering significantly enhances the ability of 2D detectors to recognize objects in the image. This narrowing of the gap is strongly supported by experiments on outdoor datasets STPLS3D, where sparse point clouds with scene-level rendering also yield good results with 2D open-world models.
>
> The Local Enforced Lookup is arranged in the appendix due to space constraints. We prioritise presenting the complete flow of the Mask-Snap-Lookup module, which is proved to be effective and is what distinguishes OpenIns3D.
>
> With these explanations, we hope the reviewer could appreciate the merits of OpenIns3D a bit better. We will address other comments in the next response. Thanks

---

> > ### Comment · Reviewer_wuwY · 2023-11-15
> >
> > I'll try to quote parts of your comments and respond inline.
> >
> > ---
> >
> > *Following this line of reasoning, it implies that the majority of point cloud processing methods, ranging from PointNet to PointTransformer, SpareNet, and Mask3D, which solely utilize 3D RGB point clouds as input without incorporating 2D images, do not offer a clear advantage over methods that demands both modalities as input.*
> >
> > I disagree here, these methods don't use the images, but they also do not claim that it is a major benefit not using the additional modality. This is the major difference to this submission.
> >
> > ---
> >
> > - *LiDAR point cloud often exists without matched 2D images. Example: Semantic3D, Tornoto3D, Paris-carla-3d, etc.*
> > - *Photogrammetry point cloud often exists without depth maps. [without depth maps it is hard to build point-pixel correspondence] Examples: Sensaturban, Campus3d, STPLS3D, etc.*
> > - *Point cloud generated from the registration of multiple scans or converted from 3D simulations/CAD models typically lacks 2D images. Example: SynthCity, etc.*
> >
> > If a colored point cloud exists, it's either synthetic, which means you could in principle create 2D images too, or there were 2D images at some point (and they might be lost). I feel it's important here to make clear that one might consider storage constraints or wants to work with old datasets, but if storage is no issue, there is no realistic scenario where 2D images don't exist, but a colored point cloud does. So I strongly dislike how this is argued in the paper, as well as in this comment, indicating that in any practical scenario colored point clouds without 2D images are realistic. I can follow along with the storage space argument, but that is not the argument in the paper.
> >
> > ---
> >
> > *Substantial Gains*
> >
> > As pointed out before, I'd like to see a ScanNet200 comparison. If the approach works as advertised, this should be super fast to compute.
> >
> > ---
> >
> > *New Insights*
> >
> > I agree that those insights are somewhat interesting, but you write a paper about a whole "mask-snap-lookup" pipeline where most components are not very novel or interesting and then move the actual interesting and potentially novel contributions to the supplementary material. Would this paper have been written in a better structure with the same results, I might agree that it should be accepted. But in my opinion, accepting the paper in the current state, signals that the whole pipeline is indeed good and/or novel and warrants publications, whereas what is actually interesting is just a specific part of the paper that is completely out of focus.
> >
> > ---
> >
> > *Gap with the natural image: Point clouds are rendered into images using ray tracing add proper lighting rather than simple point projection.*
> >
> > Where can I find this information in the paper? Did you evaluate this? Is this important?
> >
> > ---
> >
> > *The Local Enforced Lookup is arranged in the appendix due to space constraints. We prioritise presenting the complete flow of the Mask-Snap-Lookup module, which is proved to be effective and is what distinguishes OpenIns3D.*
> >
> > As pointed out above, I don't agree this is good. The core of the paper should be the main contribution. The paper could roughly outline the whole flow on a single page with a more detailed explanation in code/supplementary, while highlighting the actually interesting parts with valuable experiments. A lot of space is lost in the main paper to some nice storytelling that is not needed to highlight the value of interesting core contributions.
> >
> > ---
> >
> > So I still have large concerns with this paper. I think a lot of the claims/arguments are somewhat misleading or not precise and the structure gives weight to the wrong contributions. While I do agree that some of the evaluations in the supplementary are actually interesting, I don't think the overall paper in it's current form should be published.

---

> > > ### Author Response · Authors · 2023-11-20
> > >
> > > Reply to reviewer response:
> > >
> > > Thanks again for your reply.
> > >
> > > - Point 1: "I disagree here, these methods don't use the images, but they also do not claim that it is a major benefit not using the additional modality. This is the major difference to this submission."
> > >
> > > OpenIns3D is the first method that works effectively without relying on 2D images, and we believe, in line with all other reviewers, it is worth highlighting this aspect in the paper. Moreover, we have compared it with various methods, regardless of their input prerequisites, and have demonstrated substantial gains.
> > >
> > > If there are still any doubts, a simple question to answer is **"how can another method perform 3D instance understanding for ARKitScene LiDAR data when no 2D image is available?"** We have demonstrated this in a demo on the anonymous GitHub Repo.
> > >
> > > - Point 2: "So I strongly dislike how this is argued in the paper, as well as in this comment, indicating that in any practical scenario colored point clouds without 2D images are realistic. "
> > >
> > > We listed many publicly available datasets that contain RGB point clouds but not 2D images. We never indicated or attempted to suggest that obtaining an RGB point cloud is possible without an RGB sensor.
> > >
> > > - Point 3: Substantial Gains
> > >
> > > We have now provided results on ScanNet200.
> > >
> > > - Point 4,5,6 about paper structure:
> > >
> > > Thanks for acknowledging that the materials in the appendix are interesting. OpenIns3D is built on numerical attempts, and many insights we document in supplementary materials would indeed be helpful for the domain. We still believe, in line with all other reviewers, that Mask-Snap-Lookup should serve as the main focus of the paper.
> > >
> > > -  "I think a lot of the claims/arguments are somewhat misleading"
> > >
> > > Again, I will copy my previous response here for this comment.
> > >
> > > **Lastly, we wish to emphasize that in the composition of this paper, we have adhered to a rigorous and integral mindset. While we are eager to share the novel and encouraging findings derived from numerous attempts, we have also been transparent in acknowledging the weaknesses and limitations in our method. All the results in the paper are easily reproducible and achieve significant improvements compared to the current SOTA. We have also invested time and effort to communicate the ideas as clearly as possible. These efforts, thankfully, have been well-received and appreciated by all other reviewers.**
> > >
> > > **We recognize the subjective nature of such critiques, We would greatly appreciate it if the reviewer could provide more specific and practical suggestions for improvement. Without concrete recommendations, we believe such comments may not contribute to enhancing the paper's quality.**
> > >
> > > Thanks again for your feedback.

---

> > > > ### Author Response · Authors · 2023-11-21
> > > >
> > > > Dear Reviewer WuWY,
> > > >
> > > > We sincerely appreciate the time you dedicated to reviewing our paper. Due to the short time left before the end of discussion period, we kindly ask you to revisit our responses, where we have diligently addressed your concerns and incorporated additional experimental results as per your request. Should you have any further inquiries, please feel free to inform us.
> > > >
> > > > We would greatly appreciate your consideration in reconsidering the score, taking into account the clarifications we've provided.
> > > >
> > > > Sincerely,
> > > > Authors

---

> > > > ### Comment · Reviewer_wuwY · 2023-11-21
> > > >
> > > > - Thank you for actually providing ScanNet200 results. I sadly don't feel they are very supporting of the method though, given how the performance drops for the less common classes. To me this means the method is not truly open vocabulary, since it only works for a rather constrained set of classes.
> > > >
> > > > - Yes, OpenInst3D works without 2D images, that is fully clear, this can be highlighted. I still find the setup contrived. I see you updated the introduction w.r.t. this. I still find this section partially to be problematic though. LiDAR datasets without images cannot be used by OpenIns3D because they have no color. Just because the images are not released, does not mean they don't exist. The way the paper is written suggests that I can now use a simple point cloud and that is simply *not true*. I see the arguments about storage space/bandwidth, but these should not be mixed! *If you want to use OpenInst3D, you either need to record 2D RGB(-D) images, or use fully synthetic colored data.* This to me also renders results on ARKitScenes irrelevant, you cannot get these point clouds without RGB images!
> > > >
> > > >
> > > >
> > > > In Summary, I disagree with the structure of the paper, the interesting parts are in the supplementary material and the main paper does not give me a good understanding about why the method should be relevant or why this specific proposed method should be used. That has to be gathered from the supplementary which is still not complete, nor does the paper do a good job of pointing out which exact deeper insights can be found in there. I think that the ScanNet200 results should be the central part of the motivation, but they were only added to the supplementary, without a reference to them in the main paper. As such I will remain with my initial rating, but I somehow doubt that this will make a major difference given all the other reviews are generally more on the positive side. I still think the paper could have been way stronger with a slightly different motivation and a significantly different structure.

---

> > > > > ### Author Response · Authors · 2023-11-22
> > > > >
> > > > > We want to address some of the minor questions raised by the reviewer first. The main response to the reviewer's concerns will follow shortly.
> > > > >
> > > > > - why novel class perform better:
> > > > >
> > > > > From the experimental results, I think this might just come down to the nature of some base classes. We can see that even though the model is trained on ScanNet200, the average precision for those base classes is lower than for the novel classes. Some classes, which appear to belong to the 'base group' according to their category names, might be quite challenging to obtain. This indeed requires further investigation.
> > > > >
> > > > > - Use the original intrinsic and pose
> > > > >
> > > > > We still find this suggestion to be rather odd. Imagine you are given an RGB-3D point cloud, and your goal is not to rely on the original images but to generate new synthetic images for testing. How would you access the original pose and intrinsic information in this scenario? The suggested approach completely overlooks the challenges involved in determining effective pose and intrinsic parameters for image rendering, which are actually the real challenges of a 2D input-free method.
> > > > >
> > > > > - Generalisation
> > > > >
> > > > > The reviewer argues that Table 6 and the provided demo example may not be sufficient to demonstrate the model's generalization capability. Please refer to Table 12, which is associated with Table 6 and provides a breakdown of per-categories class results for Table 6, with a focus on the highlighted novel classes in those datasets. Many novel classes indeed prove to be rather promising with a pre-trained model trained on other dataset and other classes.
> > > > >
> > > > > It is rather challenging to determine whether the new mask in the demo belongs to the base classes or novel classes; this can only be discerned with a detailed per-categories results, which is hard to obtain as many of them has no GT. **The fact that it can be generated across datasets, proposing reasonable masks, proves that this type of mask proposal module utilized by OpenIns3D and OpenMask3D has the potential to scale up on different datasets**. We believe this is sufficient to demonstrate that it is a promising approach for instance-level understanding in 3D.

---

> > > > > > ### Author Response · Authors · 2023-11-22
> > > > > > **Concluding Points for Discussion with Reviewer wuwY**
> > > > > >
> > > > > > I would like to extend our gratitude for the reviewer's further response. We also respect the reviewer's opinion. Regarding the main disagreement, I would like to make a final argument on them and explain the changed we have made in the paper.
> > > > > >
> > > > > > # Three main disagreement
> > > > > >
> > > > > > ## 2D input free framework
> > > > > >
> > > > > > We have thoroughly reviewed our paper, addressing every sentence or phrase that might introduce any level of ambiguity regarding how the methods are applied. **The OpenIns3D setup is 2D input-free but necessitates a point cloud with RGB information. This point is crystal clear; we do not have to and will not intend to introduce even 0.1% of ambiguity on this point that might suggest otherwise**. Therefore, we disagree with the reviewer's comment that our statement on the setup is problematic.
> > > > > >
> > > > > > This still does not overshadow the fact that this setup is significantly more convenient to use than any other framework that requires well-aligned 2D images. Not to mention, on many indoor and outdoor datasets, it consistently achieves much stronger performance than SOTA even when compared to those using 2D aligned images.
> > > > > >
> > > > > > ## ScanNet200
> > > > > >
> > > > > > Compared with the majority of methods, OpenIns3D obtains results on ScanNet200 without utilizing any original 2D images; hence, the results are relatively weaker. The reason is simply that many of the common and tail classes are not clearly presented in the reconstructed 3D point cloud. **A good proof of this is the OpenScene (3D only) model. Despite being massively trained on 2D images, it still struggles to perform well on tail and common classes without referencing 2D images. Compared with this model, OpenIns3D, even without training on original 2D images, outperforms it across all category groups.**
> > > > > >
> > > > > > We acknowledge that the performance on ScanNet200 is worth investigaing and this could be considered as future work. However, dismissing all major contributions of this paper simply because it doesn't perform as well as other models that has more input information is rather unreasonable. Furthermore, please bear in mind that ScanNet200 is an experiment conducted during the rebuttal period, where not much time could be spent to further investigate the issue and enhance the results.
> > > > > >
> > > > > > ## Paper Structure
> > > > > >
> > > > > > As explained many times before, OpenIns3D aims to propose a 2D-input-free framework for 3D open-world understanding, aiming for convenience of usage as well as robust performance across datasets. At the moment, no such framework exists and has performance closer to OpenIns3D. We use the main paragraph to explain the challenges for such a model and how the proposed "Mask-Snap-Lookup" module can help. This is not failed attempts (as described by the reviewer); rather, it is the insights that this paper is trying to convey alongside the well-performed methodologies. The failed attempts, though, are documented in the appendix, Section E. We are pleased that the author finds many sections in the appendix interesting and would like to accept this submission if we include them in the main paper. However, we still the current structure is the right way, as aligned with the comments from other reviewers. Nevertheless, we respect your opinion fully and really appreciate your input on this.
> > > > > >
> > > > > > Finally, we would like to express our gratitude again for your feedback on the paper.
> > > > > >
> > > > > > Sincerely,
> > > > > > OpenIns3D team

---

> > > > > > > ### Author Response · Authors · 2023-11-23
> > > > > > >
> > > > > > > Dear reviewer wuwY,
> > > > > > >
> > > > > > > At this last point, we kindly ask the reviewer to check out the latest revision, where we made every effort to address the narrative of OpenIns3D as 2D-input-free framework, but require RGB in point cloud.
> > > > > > >
> > > > > > > Best,
> > > > > > >
> > > > > > > OpenIns3D team

---

> > ### Author Response · Authors · 2023-11-19
> >
> > Continue with the previous response:
> >
> > # Item 3 and 4
> >
> > We will first present additional experiments we conducted on ScanNet200 and then answer questions about the generalization capability of the Mask Proposal Module.
> >
> > ## Experiments on ScanNet200
> >
> > As suggested by the reviewer, we followed OpenMask3D conduct additional experiments on ScanNet200.
> >
> > Table 1: Evaluation on ScanNet200 compared with OpenMask3d and OpenScene
> >
> > |Model|Require Aligned Images in training|Require Aligned Images in Inference|Head AP|Common AP|Tail AP|AP|AP50|AP25|
> > |:-:|:-:|:-:|:-:|:-:|:-:|:-:|:-:|:-:|
> > |OpenScene(2DFusion O-Seg)|Yes|Yes|13.4|11.6|9.9|11.7|15.2|17.8|
> > |OpenScene(2D/3DEns.)|Yes|Yes|11.0|3.2|1.1|5.3|6.7|8.1|
> > |OpenScene(2DFusionL-seg)|Yes|Yes|14.5|2.5|1.1|6.0|7.7|8.5|
> > |OpenMask3D|Yes|Yes|17.1|14.1|14.9|15.4|19.9|23.1|
> > |OpenScene (3DDistill)|Yes|**No**|10.6|2.6|0.7|4.8|6.2|7.2|
> > |OpenIns3D|**No**|**No**|**16.0**|6.5|4.2|8.8|10.3|14.4
> >
> > Observation:
> > - In the inference stage, only OpenIns3D and OpenScene (3DDistill) does not require 2D aligned image as input, comparing between these two, OpenIns3D show stronger performance across all categories splits, and especially on the head categories by 5.4%.
> >
> > - Compared to other frameworks using 2D images, especially OpenMask3D, OpenIns3D exhibits weaknesses in common and tail categories. This is expected, due to the dilution or loss of visual details for small objects during the 3D reconstruction process. This limitation was clearly communicated in the original submission (Appendix E, point 3). However, in head categories, OpenIns3D demonstrates strong capabilities, even when compared with various OpenScene models.
> >
> > Table 2 Mask Understanding comparison, using Ground truth mask.
> >
> > |Models|Require2Dinputintraining|Require2Dinputininference|head(AP)|common(AP)|Tail(AP)|
> > |:-:|:-:|:-:|:-:|:-:|:-:|
> > |OpenScene(2DFusion)-OpenSeg|Yes|Yes|26.2|22|20.2|
> > |OpenScene(2DFusion)-LSeg|Yes|Yes|26.9|5.2|1.7|
> > |OpenMask3D|Yes|Yes|31.1|24|31.9|
> > |OpenIns3D|No|No|**27.3**|8.7|5.6|
> >
> > This table provides a direct comparison of mask understanding capability between OpenMask3D and OpenIns3D. Consistent with previous observations, OpenIns3D demonstrates weaknesses in common and tail classes but performs competitively in head classes, considering its more flexible input setting.
> >
> > We will add experimental results from Table 1 and Table 2 to the weaknesses section of the paper to comprehensively evaluate the performance of OpenIns3D on challenging tasks.
> >
> > Lastly, we evaluate the generalization of ScanNet20-trained Mask Proposal on ScanNet200, following OpenMask3D's base and novel settings in Table 3. The base classes are the 53 classes picked from the 200 categories that are close to the ScanNet20 classes (followed by OpenMask3D).
> >
> > Table 3: Class-agonstic mask evaluation on ScanNet200
> > |  | AP | AP50 | AP25 | AP | AP50 | AP25 | AP | AP50 | AP25 |
> > |---|---|---|---|---|---|---|---|---|---|
> > | Trained on | Overall |  |  | Base |  |  | Novel |  |  |
> > | ScanNet20 | 0.3922 | 0.5428 | 0.6371 | 0.2669 | 0.3664 | 0.4557 | **0.4035** | **0.586**| **0.6878** |
> > | ScanNet200 | 0.4805 | 0.6607 | 0.7512 | 0.3486 | 0.4777 | 0.5702 | **0.4466** | **0.6436** | **0.7544** |.
> >
> > MPM trained on ScanNet20, although it has slightly worse performance than MPM trained on ScanNet200, still offers relatively good mask proposals. This aligns with the observation made in OpenMask3D. Interestingly, the performance gap in novel classes is smaller than that in the base classes.
> >
> > ## Generalization Capability of OpenIns3D
> >
> > We hereby provide three evidences:
> >
> > 1. In the original paper, Table 6 presents a cross-dataset analysis, where OpenIns3D is trained on one dataset while being tested on the other. The encouraging results show good performance for cross-domain zero-shot understanding. Table 11 provides detailed per-category results, highlighting those novel categories.
> >
> > 2. In Table 3 above, when the model is trained on ScanNetv2 and tested on ScanNet200, it shows decent performance in class-agnostic mask proposal on novel classes.
> >
> > 3. We conducted zero-shot experiments with OpenIns3D, trained on ScanNet, across four diverse types of indoor datasets. The visual results are presented in the following anonymous repository: (https://anonymous.4open.science/r/rebuttal_image-6CBE/README.md).
> >
> > Combining all three pieces of evidence, we believe it provides strong proof of how MPM can be generalised.
> >
> > In fact, given that OpenMask3D and OpenIns3D share a similar mask proposal module, the evidence we provided here, combined with evidence from OpenMask3D, does demonstrate that this type of query-based mask proposal module has good scalability in 3D. Similar backbones have been proven successful in 2D, as demonstrated by models like Mask2Former and SAM, especially when trained with large-scale datasets. We believe this conveys an important message.
> >
> > We will be addressing the remaining items, as well as your additional responses, shortly.

---

> > > ### Author Response · Authors · 2023-11-20
> > >
> > > Continue with the previous response:
> > >
> > > ## Item 5 Using original image's pose and intrinsic
> > >
> > > Thank you for the suggestion; however, could the reviewer kindly elaborate on the intention behind this setting? **In scenarios where 2D images are not accessible—how could we get to know the pose and intrinsic parameters of those images?** Through our numerous attempts with a 2D image-free approach, we've discovered that one of the main challenges is how to determine effective and efficient pose and intrinsic. Directly taking pose and intrinsic from the original 2D image might not be so logical. OpenMask3D demonstrates a similar setup in the paper in its latest revision, but we are unsure the intention on this.
> > >
> > > ## Item 6 3D DETR
> > >
> > > The concern regarding the use of pre-trained 3D models lies in their tendency to bypass the challenges associated with proposing bounding boxes, which are crucial tasks in 3D scene-level understanding.
> > >
> > > It's important to emphasize that there is a lack of widely adopted 3D pre-trained models. To our knowledge, in their context, a 3D pre-trained model generally refers to being trained on the same dataset used for testing, supervised by all ground truth labels and classes during training. In fact, this practice is commonplace in the field of 3D visual grounding, because all benchmarks in that domain, are originated from the same dataset, ScanNetv2. However, we think this is not a good practice.
> > >
> > > Several works in the field, such as the PLA family, Openmask3D, OpenScene, and OpenIns3D, demonstrate a dedicated focus on the challenges of 3D **scene-level understanding**. What sets these works apart is their simultaneous consideration of both proposal and understanding challenges. Therefore, we believe it is crucial to distinguish between works that solely rely on pre-trained box proposals and those that address both aspects.
> > >
> > > However, we agree that using 2D DETR is reasonable, as they normally have good generalisation capability. We will modify the text accordingly.
> > >
> > > ## Last point
> > >
> > > **Lastly, we wish to emphasize that in the composition of this paper, we have adhered to a rigorous and integral mindset. While we are eager to share the novel and encouraging findings derived from numerous attempts, we have also been transparent in acknowledging the weaknesses and limitations in our method. All the results in the paper are easily reproducible and achieve significant improvements compared to the current SOTA. We have also invested time and effort to communicate the ideas as clearly as possible. These efforts, thankfully, have been well-received and appreciated by all other reviewers.**
> > >
> > > **Regarding the reviewer's observation that our paper may contain a lot of "fluff," introducing new fancy names for various concepts, we recognize the subjective nature of such critiques. We would greatly appreciate it if the reviewer could provide more specific and practical suggestions for improvement. Without concrete recommendations, we believe such comments may not significantly contribute to enhancing the paper's quality.**
> > >
> > > # Reply to all questions
> > >
> > > 1. Thanks for point this out, we will update this in new revision.
> > >
> > > 2. There are two layers of argument here that we would like to convey, and the second argument is more important. 1) Yes, it is possible to project and get depth, but it is rather inconvenient 2) In our experience with various 2D-3D dataset alignments, it's highly probable that a depth map generated in such scenarios will lack accuracy and may not be very useful. **The precision of 2D-3D alignment heavily relies on the original execution of 3D reconstruction.** In datasets like ScanNet, where the reconstruction is notably good, there are only minor errors in the depth map and pose information. On the flip side, datasets like ArkitScene exhibit a much higher level of inaccuracy in 2D-3D alignment, making it challenging to link both modalities seamlessly. This introduces additional challenges for approaches that rely on both types of data. **While the assumption that directly projecting points to create a depth map is valid in the presence of perfect 2D-3D alignment, in reality, achieving such perfection is rarely the case.**
> > >
> > > 3. "As suggested above, an actual evaluation on ScanNet200 would make a major difference."
> > > As suggested, we have now included ScanNet200 results. Please let us know if your concerns have been addressed.
> > >
> > > We would like to thank the reviewer again for their detailed feedback and suggestions on enhancing the paper.

---

> > > > ### Comment · Reviewer_wuwY · 2023-11-21
> > > >
> > > > # Using original image's pose and intrinsic
> > > > For the sake of ablation studies, I don't care about scenarios where 2D images are not available. For ScanNet it would have been easy to compare OpenIns3D with existing camera intrinsics. You focused on object specific views and views from around and above the scene, however, the third and to me very natural option would be views from inside the scene as used during data recording. This would have given you the option to directly compare how well your rendering compares to the original images.
> > > >
> > > > #  Pretrained 3D DETR
> > > > Still for most of the experiments you use a pretrained Mask3D model for the proposals, trained on ScanNet, evaluated on ScanNet, so in essence you do the same thing. I see that you took out the sentence about training from scratch though, so this point is indeed solved.
> > > >
> > > > # Last point
> > > > Every paper out there probably has a long story to it, with many failed attempts. If you want to tell them, tell them in a blog post, or a talk, or in the *appendix*. I strongly disagree that the front and center part of the paper should be used for this narrative. A lot of people will read a paper without the supplementary, especially when the supplementary is so long as the one of this paper. The story that this main paper tells, is simply not convincing to me. The masking module is not new, the way the main paper introduces the lookup module is extremely simplistic and suggests there is nothing new to it, and only the snap part feels somewhat interesting, but again, the only interesting stuff to me is in the supplementary. Furthermore, information about how you actually render the images, which feels very crucial, are not present at all.

---

> > > ### Comment · Reviewer_wuwY · 2023-11-21
> > >
> > > # ScanNet200 Results
> > > The ScanNet200 results for me clearly indicate OpenIns3D has a big issue with generalizing to the tail common and tail classes. While OpenMask3D is by far not perfect, its performance drops significantly less. This should be the core of any open vocabulary method. So I don't find these numbers very convincing. And if there is a clear weakness to a system, I don't think it's good to point this out somewhere deeply hidden in a huge appendix. Regardless of that, just because something is a tail class, doesn't obviously make it a small class, so this weakness would have to be investigated more closely.
> > >
> > > I'm also somewhat confused by the fact that Table 3 in the above comment lists higher performances for the novel classes than for the base classes. How is that explained?
> > >
> > > # Generalization
> > > Indeed Table 3 suggests that Mask3D can generalize, but as I pointed out above, I am very confused by the fact that it works better on novel classes. This is very much not in line with the overall degrading performance on novel classes in the tables before. The additional visualizations are beside the point here, given that this just shows that Mask3D can generalize somewhat across datasets, but it doesn't indicate how well it is able to propose class agnostic masks for new classes. (Which is also the case for Table 6)
> > >
> > > I also disagree with your points about Mask2Former and SAM. I am not aware of class agnostic Mask2Formers and SAM was trained in a very different way altogether. So the only real evidence for generalization to new classes is table 3 and I'd like to hear your comment on why it actually works better for novel classes.

---

### Official Review · Reviewer_rEPE · 2023-11-03

**Soundness:** 3 good
**Presentation:** 2 fair
**Contribution:** 3 good
**Rating:** 6
**Confidence:** 3

**Summary:**

OpenIns3D, a RGB-independent framework, addresses point cloud-based instance segmentation through the introduction of a multiple stages approach. Firstly, it generates mask proposals based on point clouds, rendering scene images following a specified strategy. Subsequently, 2D vision language models are employed to extract objects. Finally, the assignment process between 2D and 3D segments is applied. Experimental results underscore that OpenIns3D substantially outperforms previous baseline models.

**Strengths:**

- The proposed framework focuses on an RGB-agnostic setting and achieves precise 3D instance segmentation through a multi-stage approach.

- The framework primarily relies on 3D proposals, establishing connections between 2D and 3D segments, and subsequently employs filtering operations that effectively leverage large-scale 2D vision models.

- Experimental results, when compared to those presented in previous papers, clearly illustrate a remarkable enhancement in performance.

**Weaknesses:**

- The Mask Proposal Module is trainable using IoU as a form of supervision. It is strongly recommended to include comprehensive training details in the main draft of the paper.
- Further clarification is needed regarding the adjustment of camera parameters in the Camera Intrinsic Calibration process.
- It is advisable to incorporate a comparative analysis that includes segmentation results obtained from multiple pseudo-projected images. Given that your method heavily relies on prior knowledge from 2D models, solely comparing it with point-based methods may not provide a fully equitable evaluation.
- The performance gain from 2D models or framework design should be separately discussed.

**Questions:**

- Please provide a detailed explanation of the several modules proposed in the paper, especially in addressing Weaknesses 1 and 2.

- Please include further comparisons with image-based segmentation. This can be achieved by projecting the point cloud onto different cameras and then unprojecting it back to the initial point clouds. Such additional analysis would help demonstrate the performance of the proposed method and offer valuable insights into why it may perform better or worse compared to pure image-based segmentation, and the gain from 2D models.

---

> ### Author Response · Authors · 2023-11-17
>
> We would like to express our gratitude for the detailed review and your support of our work. We believe that OpenIns3D, as the first 2D-input-free framework for 3D open-world scene understanding, will not only set new benchmark results but also offer a fresh perspective to the domain. Our response:
>
> # More training details of Mask scoring & Intrinsic parameter calibration:
> Thanks for your suggestion. We have explained the details as follows and will include this in the revision (will update later).
> - Mask scoring: For each of the $N$ proposed masks, we utilized the Hungarian Match to match it with $n$ GT masks and calculated the IOU value between the each pairs (n). For the remaining masks $(N-n)$ that did not match with GT, we set the IoU value as zeros. With this method, a $N$ GT IoUs for all proposed masks are obtained. During training, a simple two-layer MLP is applied to process mask queries and predict their IoU value. This simple two-layer MLP is supervised by the $L_2$ loss between predicted IoU and GT IoU, as depicted in formula (1) in the main paper.
>
> - Intrinsic parameter calibration:
> Once the pose matrix $Pose$ is obtained via the $Lookat$ function (A.2 in appx), we initialize the Intrinsic with an ordinary intrinsic matrix. With this parameter, we project all points into the images. This will lead to randomly positioned scene images. Once this is obtained, we uniformly rescale the values of fx, fy, cx, and cy in the initialised intrinsic matrix by the same factor. This process merely repositions and rescales the projected image. To illustrate, if the original projected point was located in image coordinates within the range of [-1000, -192] in x, our calibrated intrinsic metrics transform it to [0, 2000] in x.
>
> We will update these parts in the the main paper later. Thanks again for the suggestion.
>
> #  Further Comparison with Image-based segmentation
>
>  We will answer Weakness 3 and Question 2 in this section as they are closely related
>
> - Comparing with non-point-based methods:
> In general, most 3D open worlds rely on 2D images, making it challenging to categorize them as truly point-based. If considered solely from an inference perspective, PLA-family and OpenScene might be point-based methods, although both heavily rely on 2D images during training. Methods like PointClip, PointClipV2, and OV-3DET are derived from multiple projections/croppings, and their comparisons are included in the paper. We present these comparisons in Table 7 (OVOD), Table 3 (outdoor dataset), ablation studies on Table 2, and pre-class results on Tables 10 and 11.
>
> -	Pros and Cons when comparing with pure-image projection-based methods:  For various pseudo-image projection methods, we showcase six attempts we made in Table 12 and compare them based on runtime and performance. Our scene-level rendering also proves to be robust in both performance and fast inference time. The downside, though, as we stated in the weakness session [point 3], is that it might not work well on small objects, as the visual information of small objects is diluted or lost in the 3D reconstruction process. For a more detailed analysis of this point, we compared our method with OpenMask3D on ScanNet200, as shown in the table below. OpenIns3D achieves competitive performance in the Head category (top 1/3 classes, generally medium size object) but shows weaknesses in common and tail categories. OpenMask3D achieves strong results in common and tail categories, at the cost of requiring well-aligned 2D image as input during inference. Compared with OpenScene, **in cases where only 3D is available during inference, OpenIns3D surpasses OpenScene in all categories.**
>
> |Model|Require Aligned Images in training|Require Aligned Images in Inference|Head AP|Common AP|Tail AP|AP|AP50|AP25|
> |:-:|:-:|:-:|:-:|:-:|:-:|:-:|:-:|:-:|
> |OpenScene(2DFusion O-Seg)|Yes|Yes|13.4|11.6|9.9|11.7|15.2|17.8|
> |OpenScene(2D/3DEns.)|Yes|Yes|11.0|3.2|1.1|5.3|6.7|8.1|
> |OpenScene(2DFusionL-seg)|Yes|Yes|14.5|2.5|1.1|6.0|7.7|8.5|
> |OpenMask3D|Yes|Yes|17.1|14.1|14.9|15.4|19.9|23.1|
> |OpenScene (3DDistill)|Yes|**No**|10.6|2.6|0.7|4.8|6.2|7.2|
> |OpenIns3D|**No**|**No**|**16.0**|6.5|4.2|8.8|10.3|14.4
>
> # performance gain from 2D models
>
> We conducted ablation experiments on various 2D models and evaluated their performance on the ScanNet20 dataset. This is included in Table 2. The following is a breakdown version of it. Notably, when we switched the rendering method to scene-level rendering, there is a significant increase in performance. This effect underscores the usefulness of scene-level rendering for 3D open-world understanding.
> |Methods|Rendering|2Dbackbone|AP50|AP25|
> |-|-|:-:|:-:|:-:|
> |Mask-P-CLIP|Mask|CLIP|4.5|14.4|
> |OpenIns3D|Scene-level|Grounding-DINO|18.8(+14.4)|29.8(+15.4)|
> |OpenIns3D|Scene-level|ODISE|28.7|38.9|
>
> We hope the rebuttal is helpful. Please let us know if you need any further clarification or have additional questions. Thanks again for your support.

---

> > ### Comment · Reviewer_rEPE · 2023-12-02
> > **Feedbacks from the reviewer**
> >
> > After reading the responses from the authors, I will keep my original rating.

---

### Author Response · Authors · 2023-11-21
**Revision Summary**

We extend our sincere gratitude to all reviewers for their valuable and constructive feedback, which has significantly enhanced the quality of our paper. In response to their suggestions, we have undertaken additional experiments and made several revisions, all of which are highlighted in blue in the PDF file. Below is a summary of these revisions:

- We changed the narrative of why 2D images may not be available in the introduction section to avoid confusion and misunderstanding (Page 2).
- We removed the emphasis that OpenIns3D is trained from scratch to avoid confusion and misunderstanding (Page 8).
- We added a new reference for CLIP^2, which categorizes in the CLIP and projection group on page 13.
- We added further explanation of the mask scoring module on page 14.
- We added a detailed explanation of the calibration of the intrinsic matrix on page 14.
- We added a new section for the experiment results on ScanNet200 on page 16, made comparisons with OpenMask3D, and discussed the pros and cons of both methods.
- We further explained the performance gain coming from the change of the 2D model on page 20.
- We cross-referenced the experiment results of ScanNet200 in the limitation and future work section.
- Cited several important references previously omitted in the original manuscript.
- Corrected various unclear expressions and grammatical errors.

---

### Author Response · Authors · 2023-11-23
**Summary of Rebuttal**

With the end time of the discussion window approaching, I would like to take this chance to thank all reviewers again for their thoughtful feedback. At the moment, we have a majority of positive reviews with one reject review, which makes this paper a borderline case. However, we believe OpenIns3D is an important work for the 3D open-world scene understanding domain and want to take this opportunity to highlight the merits of OpenIns3D and how it can benefit the domain.

1. **First 2D-input-free framework for scene-level understanding.** While most works on 3D open-world scene understanding demand posed 2D images, depth maps, and point clouds as input, OpenIns3D distinguishes itself by solely requiring coloured 3D point clouds for both training and inference. Such a framework is new to the domain and can be utilized in numerous scenarios where other models are not applicable.

2. **Significant OVOD Task Improvement over Latest Papers.** Within the current literature on 3D open-world object detection, a prevalent trend is observed where even papers accepted at leading conferences such as CVPR/ICCV/NIPS, along with some ICLR submissions, consistently report ScanNet results around 20%. In contrast, OpenIns3D can easily achieve $AP_{25}$ range of 35-45%. This underscores the urgency of our work at this juncture. With our work, we wish to signal that achieving results within the 20% range on ScanNet is insufficient, and the pursuit of incremental improvements in this task may no longer be as beneficial or worthwhile.

3. **SOTA OVIS performance on indoor/outdoor & Zero-shot cross-domain.** Our results on S3DIS are 17% higher than the newly extended work (Lowis3D) of the previous SOTA (PLA). We also demonstrate performance on the sparse outdoor point cloud dataset STPLS3D( + >10%), proving the effectiveness of our approach in varied settings.  Upon request, we also showcase how OpenIns3D can be easily deployed and applied to other datasets for zero-shot understanding on this page  (https://anonymous.4open.science/r/rebuttal_image-6CBE/README.md). This includes Lidar-based (ArkitScene) point clouds with no images available, where other models are not applicable. In a rapidly advancing field, I believe these encouraging results hold substantial value for the domain and can contribute to pushing the boundaries forward.

We want to emphasize that none of the above statements is overstated. Our results are easily reproducible and we believe that any proper implementation of OpenIns3D can easily surpass the current state-of-the-art on the aforementioned benchmarks by a large margin.

4. **Complex input queries handling** Other than that strong numerical improvement, OpenIns3D can effectively handle complex language queries when integrated with the LLM-powered 2D model. This makes it able to understand long sentences, queries that require world knowledge, complex reasoning, and carry out segmentation tasks. **To our knowledge, no other 3D open-world scene understanding model can do so and process this level of free-flowing, reasoning-demanding textual queries**.

5. **New insights.** Valuable insights on rendering techniques: OpenIns3D introduces a scene-level rendering and matching scheme that is significantly faster and more effective than all existing single-mask rendering/cropping schemes. Even in the case of Reviewer wuwY, who proposed rejection, stating that they find these insights rather interesting, they would consider acceptance with the current results if we were to restructure the paper (although we disagree with this suggestion).

We hope this information will be helpful for further discussion and instils confidence in our work. We sincerely appreciate all the input from the reviewers, which has significantly improved the quality of the paper.

Best,

OpenIns3D team

---

### Author Response · Authors · 2023-11-23
**Summary of the conversation with Reviewer wuwY**

Dear reviewers and area chair,

At the moment, Reviewer wuwY is the only one holding a negative rate of our submission. There have been several back-and-forth conversations with Reviewer wuwY, with many concerns being addressed or proven untrue. We believe it will be helpful to provide a concise summary of the remaining disagreements for your reference.

- 3D-input only

The reviewer expressed a dislike for the narrative of the 3D-only setup of our model, suggesting that it implies compatibility with point clouds without RGB. We understand the concerns and have made every effort to articulate this point clearly throughout the paper. **We have thoroughly reviewed the paper, addressing any sentence or phrase that might introduce any level of ambiguity on this point. Therefore, we believe this concern has been fully addressed.** The setup of OpenIns3D needs no aligned 2D images but does require point clouds with colours. This is clear and it is far more convenient to use in practice, let alone the robust performance it achieved across indoor and outdoor datasets.

- Scannet200 results

Upon the reviewer's request, we conducted a quick experiment on ScanNet200. The results show that OpenIns3D demonstrates predominantly better performance on all category groups over models that only use 3D input, but exhibits weaker performance compared to OpenMask3D, which heavily references original well-aligned 2D images during inference.

We believe we have explained the reason very clearly: this is because many common and tail classes are not clearly presented in reconstructed 3D. We agree with the reviewer that this is worth further investigation and could be useful for future work. However, overshadowing all the merits of OpenIns3D just because of this aspect is rather unreasonable. Not to mention:

1.	**OpenMask3D was published after the submission deadline for ICLR, with the latest version on arXiv published on 29 Oct 2023**. Making such strong criticism of our submission based on this might be rather harsh and unfair (not sure if this is allowed in ICLR??)

2.	OpenMask3D utilizes 2D images, which naturally provide stronger performance when it comes to details on tail classes.

3.	Experiments on ScanNet200 were conducted during the rebuttal phase, with limited time for in-depth investigation and improvements.

**We must sadly say that putting a novel submission with many unique merits done, solely because it cannot outperform a concurrent work, published one month after the deadline, in one dataset, with an unequal evaluation, is rather non-inclusive and discouraging.** Such a practice may not be beneficial for the long-term progress of this early-stage, rapidly evolving, and important domain.

- Paper structure

Lastly, the author suggests that the paper's structure is not right. While we respect this opinion, considering all other reviewer comments, it seems clear that this is not a mainstream viewpoint.

We cite the reviewer's opinion here:

> “Would this paper have been written in a better structure with the same results, I might agree that it should be accepted. But in my opinion, accepting the paper in the current state, signals that the whole pipeline "mask-snap-lookup" is indeed good and/or novel and warrants publications, whereas what is actually interesting is just a specific part of the paper that is completely out of focus.”

**We are pleased that the reviewer found many of the details documented in the supplementary material interesting. However, we must emphasize that "mask-snap-lookup" is indeed crucial. Without it, the model cannot perform in the 2D-input-free setting, cannot achieve the SOTA results across datasets, and cannot effectively handle complex input queries, cannot deliver new insights of scene-level rendering.** Our viewpoint is consistent with the opinions of all other reviewers. Nevertheless, we respect your opinion, as you may have your own stance.

Assigning a "Score 3" indicates a strong negative evaluation, but we strongly believe that the merits of OpenIns3D, may not have been fully appreciated by reviewer wuwY. We hope this perspective is taken into account.

**With that said, we genuinely appreciate reviewer wuwY for actively engaging in the discussion and kindly request a reconsideration of the score. If not, we hope this comment will be useful for further discussions.**

Sincerely,

OpenIns3D team

---

### Meta-Review · Area_Chair_zGLm · 2023-12-10

**Metareview:**

**Summary**
The paper proposes a "mask-snap-lookup" framework for 3D open-vocabulary instance segmentation on colored point clouds that does not require any 2D image inputs ("2D input free").  A mask-proposal-module (MPM) based on Mask3D is trained to proposes class-agnostic "masks" (on the point cloud) with IOU scores (the scores are used to help filter out bad proposals).  Contrary to the claim that the method is "2D input-free", the method actually does use 2D images, but these are rendered 2D images vs "input" 2D images (e.g. high-quality RGB camera images) that are provided as input in addition to the point cloud.  In the the "snap" stage, these 2D images are rendered by selecting appropriate scene-level viewpoints that point inward toward the scene.  A 2D open-world detector is then used on the rendered scene images to obtain possible object classes which are stored in "lookup" tables.   During the "lookup" stage, the 3D mask proposals are projected onto the 2D image planes and used with the class lookup tables (from 2D detectors) to determine object categories.  The framework is evaluated on both indoor and outdoor point cloud datasets and shown to outperform prior work.

The main contribution of this work is the proposed "mask-snap-lookup" framework.  The paper emphasizes also the use of rendered images of the point cloud vs needing to have aligned 2D-images provided as input.

**Strengths**
- The proposed framework is flexible
- Experiments show the proposed method outperforms prior work
- Experiments are done with seen and unseen classes

**Weaknesses**
- Reviewers noted missing details about different components.  The AC also found many details hard to piece together.  It is not clear whether sufficient detail about the method is provided in the revised draft
- Reviewer wuwY raised questions about what is the main novel components
- Experiments in the main paper did not really show performance on unseen categories.
- Overall, experiments does not really evaluate the open-vocabulary capabilities of the model as evaluation is mostly based on limited number of categories.

**Recommendation**

The AC believes that there is value in this work.  However, based on reviewer feedback, many important details seems unclear and the structure of the paper should be improved.  As revisions are necessary before the work is ready for ICLR, the AC recommends reject.


**Suggestions for improvement**

The AC's main concern is with the clarity of the writing, and whether important details are provided in the main paper, the authors are recommended to improve the writing and experimental details so that it is easier for reviewers to judge the claims of the work.

In particular, the AC would recommend
- Tone down the claim that it is "2D input free".  Overall, the claim that it is "2D input free" while relying on applying 2D detectors on rendered 2D images is somewhat contradictory and confusing.  The AC recommends that the writing be improved so what exactly is meant by "2D input free" is clearly defined.  Based on the paper, it seems that the authors use the term "2D input" to mean images that were captured by a high-resolution RGB camera, while the term "synthetic image" is used to refer the images rendered from the 3D point cloud.  This use of the term "synthetic image" is not necessarily well-accepted and should be clarified.  Currently, most of the information about how the "synthetic image" is obtained is found in the supplement, and it is not that clear from the main paper that the "synthetic image" is rendering from the point-cloud.  As this appears to be a main point of the paper (that only uses the colored point cloud and not extra 2D image directly captured by the camera or rendered via a mesh), this should be clarified and discussed more in the main paper.

- "Snap": Explain clearly the intuition for the camera positioning and setting of focal length and central coordinates and how it addresses challenges 2,3.  Note that using rendered images for 3D scene understanding is not uncommon.

- "Lookup": The "Local Enforced Lookup" seems to be the main technical novel component.  Please explain it more in the main paper.

- Figure 7: This looks to be a good illustration of how your work is different from prior work.  It should go the main paper.

- If your focus on "open-vocabulary", have more "open-vocabulary experiments in the main paper.  Close-set experiments and cross-domain ablations can be moved to the supplement.  Try to go beyond evaluating on a small fix set of classes (even if some of them are "novel") if you want to tackle "open-vocabulary".

**Justification For Why Not Higher Score:**

While the paper proposes an interesting framework, the paper structure makes it difficult to identify the novel components and their details.  In addition, there are limited experiments showing the open-vocabulary ability of the model.  There are some experiments with limited number of unseen classes (most of discussion on performance on unseen classes are actually in the supplement), but there are not true open-vocabulary experiments (other than qualitative Figure 6).

**Justification For Why Not Lower Score:**

N/A

---

### Decision · Program_Chairs · 2024-01-16

Reject